# Reinforcement Learning with Perturbed Rewards

## Abstract

Recent studies have shown that reinforcement learning (RL) models can be vulnerable in various scenarios, where noises from different sources could appear. For instance, the observed reward channel is often subject to noise in practice (e.g., when observed rewards are collected through sensors), and thus observed rewards may not be credible. Also, in applications such as robotics, a deep reinforcement learning (DRL) algorithm can be manipulated to produce arbitrary errors. In this paper, we consider noisy RL problems where observed rewards by RL agents are generated with a reward confusion matrix. We call such observed rewards as *perturbed rewards*. We develop an unbiased reward estimator aided robust RL framework that enables RL agents to learn in noisy environments while observing only perturbed rewards. Our framework draws upon approaches for supervised learning with noisy data. The core ideas of our solution include estimating a reward confusion matrix and defining a set of unbiased surrogate rewards. We prove the convergence and sample complexity of our approach. Extensive experiments on different DRL platforms show that policies based on our estimated surrogate reward can achieve higher expected rewards, and converge faster than existing baselines. For instance, the state-of-the-art PPO algorithm is able to obtain 67.5% and 46.7% improvements in average on five Atari games, when the error rates are 10% and 30% respectively.

## 1 Introduction

Designing a suitable reward function plays a critical role in building reinforcement learning models for real-world applications. Ideally, one would want to customize reward functions to achieve application-specific goals (Hadfield-Menell et al., 2017). In practice, however, it is difficult to design a function that produces credible rewards in the presence of noise. This is because the output from any reward function is subject to multiple kinds of randomness:

- *Inherent Noise*. For instance, sensors on a robot will be affected by physical conditions such as temperature and lighting, and therefore will report back noisy observed rewards.

- *Application-Specific Noise*. In machine teaching tasks (Thomaz et al., 2006; Loftin et al., 2014), when an RL agent receives feedback/instructions from people, different human instructors might provide drastically different feedback due to their personal styles and capabilities. This way the RL agent (machine) will obtain reward with bias.

- *Adversarial Noise*. Adversarial perturbation has been widely explored in different learning tasks and shows strong attack power against different machine learning models. For instance, Huang et al. (2017) has shown that by adding adversarial perturbation to each frame of the game, they can mislead RL policies arbitrarily.

Assuming an arbitrary noise model makes solving this noisy RL problem extremely challenging. Instead, we focus on a specific noisy reward model which we call *perturbed rewards*, where the observed rewards by RL agents are generated according to a reward confusion matrix. This is not a very restrictive setting to start with, even considering that the noise could be adversarial: Given that arbitrary pixel value manipulation attack in RL is not very practical, adversaries in the real-world have high incentives to inject adversarial perturbation to the reward value by slightly modifying it. For instance, adversaries can manipulate sensors via reversing the reward value.

In this paper, we develop an unbiased reward estimator aided robust framework that enables an RL agent to learn in a noisy environment with observing only perturbed rewards. Our solution framework builds on existing reinforcement learning algorithms, including the recently developed DRL ones ($Q$-Learning (Watkins, 1989; Watkins & Dayan, 1992), Cross-Entropy Method (CEM) (Szita & Lörincz, 2006), Deep SARSA (Sutton & Barto, 1998), Deep $Q$-Network (DQN) (Mnih et al., 2013; 2015; van Hasselt et al., 2016), Dueling DQN (DDQN) (Wang et al., 2016), Deep Deterministic Policy Gradient (DDPG) (Lillicrap et al., 2015), Continuous DQN (NAF) (Gu et al., 2016) and Proximal Policy Optimization (PPO) (Schulman et al., 2017) algorithms).

The main challenge is that the observed rewards are likely to be biased, and in RL or DRL the accumulated errors could amplify the reward estimation error over time. We do not require any assumption on knowing the true distribution of reward or adversarial strategies, other than the fact that the generation of noises follow an unknown reward confusion matrix. Instead, we address the issue of estimating the reward confusion matrices by proposing an efficient and flexible estimation module. Everitt et al. (2017) provided preliminary studies for the noisy reward problem and gave some general negative results. The authors proved a *No Free Lunch* theorem, which is, without any assumption about what the reward corruption is, all agents can be misled. Our results do not contradict with the results therein, as we consider a specific noise generation model (that leads to a set of perturbed rewards). We analyze the convergence and sample complexity for the policy trained based on our proposed method using surrogate rewards in RL, using $Q$-Learning as an example.

We conduct extensive experiments on OpenAI Gym (Brockman et al., 2016) (AirRaid, Alien, Carnival, MsPacman, Pong, Phoenix, Seaquest) and show that the proposed reward robust RL method achieves comparable performance with the policy trained using the true rewards. In some cases, our method even achieves higher cumulative reward - this is surprising to us at first, but we conjecture that the inserted noise together with our noisy-removal unbiased estimator adds another layer of exploration, which proves to be beneficial in some settings. This merits a future study.

Our contributions are summarized as follows: (1) We adapt and generalize the idea of defining a simple but effective unbiased estimator for true rewards using observed and perturbed rewards to the reinforcement learning setting. The proposed estimator helps guarantee the convergence to the optimal policy even when the RL agents only have noisy observations of the rewards. (2) We analyze the convergence to the optimal policy and finite sample complexity of our reward robust RL methods, using $Q$-Learning as the running example. (3) Extensive experiments on OpenAI Gym show that our proposed algorithms perform robustly even at high noise rates.

## 1.1 RELATED WORK

**Robust Reinforcement Learning**   It is known that RL algorithms are vulnerable to noisy environments (Irpan, 2018). Recent studies (Huang et al., 2017; Kos & Song, 2017; Lin et al., 2017) show that learned RL policies can be easily misled with small perturbations in observations. The presence of noise is very common in real-world environments, especially in robotics-relevant applications. Consequently, robust (adversarial) reinforcement learning (RRL/RARL) algorithms have been widely studied, aiming to train a robust policy which is capable of withstanding perturbed observations (Teh et al., 2017; Pinto et al., 2017; Gu et al., 2018) or transferring to unseen environments (Rajeswaran et al., 2016; Fu et al., 2017). However, these robust RL algorithms mainly focus on noisy vision observations, instead of the observed rewards. A couple of recent works (Lim et al., 2016; Roy et al., 2017) have also looked into a rather parallel question of training robust RL algorithms with uncertainty in models.

**Learning with Noisy Data**   Learning appropriately with biased data has received quite a bit of attention in recent machine learning studies Natarajan et al. (2013); Scott et al. (2013); Scott (2015); Sukhbaatar & Fergus (2014); van Rooyen & Williamson (2015); Menon et al. (2015). The idea of above line of works is to define unbiased surrogate loss function to recover the true loss using the knowledge of the noises. We adapt these approaches to reinforcement learning. Though intuitively the idea should apply in our RL settings, our work is the first one to formally establish this extension both theoretically and empirically. Our quantitative understandings will provide practical insights when implementing reinforcement learning algorithms in noisy environments.

## 2 PROBLEM FORMULATION AND PRELIMINARIES

In this section, we define our problem of learning from perturbed rewards in reinforcement learning. Throughout this paper, we will use *perturbed reward* and *noisy reward* interchangeably, as each time step of our sequential decision making setting is similar to the "learning with noisy data" setting in supervised learning (Natarajan et al., 2013; Scott et al., 2013; Scott, 2015; Sukhbaatar & Fergus, 2014). In what follows, we formulate our Markov Decision Process (MDP) problem and the reinforcement learning (RL) problem with perturbed (noisy) rewards.

### 2.1 REINFORCEMENT LEARNING: THE NOISE-FREE SETTING

Our RL agent interacts with an unknown environment and attempts to maximize the total of his collected reward. The environment is formalized as a Markov Decision Process (MDP), denoting as $\mathcal{M} = \langle \mathcal{S}, \mathcal{A}, \mathcal{R}, \mathcal{P}, \gamma \rangle$. At each time $t$, the agent in state $s_t \in \mathcal{S}$ takes an action $a_t \in \mathcal{A}$, which returns a reward $r(s_t, a_t, s_{t+1}) \in \mathcal{R}$ (which we will also shorthand as $r_t$), and leads to the next state $s_{t+1} \in \mathcal{S}$ according to a transition probability kernel $\mathcal{P}$, which encodes the probability $\mathbb{P}_a(s_t, s_{t+1})$. Commonly $\mathcal{P}$ is unknown to the agent. The agent's goal is to learn the optimal policy, a conditional distribution $\pi(a|s)$ that maximizes the state's value function. The value function calculates the cumulative reward the agent is expected to receive given he would follow the current policy $\pi$ after observing the current state $s_t$: $V^\pi(s) = \mathbb{E}_\pi \left[ \sum_{k=1}^\infty \gamma^k r_{t+k} \mid s_t = s \right]$, where $0 \leq \gamma \leq 1$[1] is a discount factor. Intuitively, the agent evaluates how preferable each state is given the current policy. From the Bellman Equation, the optimal value function is given by $V^*(s) = \max_{a \in \mathcal{A}} \sum_{s_{t+1} \in \mathcal{S}} \mathbb{P}_a(s_t, s_{t+1}) \left[ r_t + \gamma V^*(s_{t+1}) \right]$. It is a standard practice for RL algorithms to learn a state-action value function, also called the $Q$-function. $Q$-function denotes the expected cumulative reward if agent chooses $a$ in the current state and follows $\pi$ thereafter: $Q^\pi(s, a) = \mathbb{E}_\pi \left[ r(s_t, a_t, s_{t+1}) + \gamma V^\pi(s_{t+1}) \mid s_t = s, a_t = a \right]$.

### 2.2 PERTURBED REWARD IN RL

In many practical settings, our RL agent does not observe the reward feedback perfectly. We consider the following MDP with perturbed reward, denoting as $\tilde{\mathcal{M}} = \langle \mathcal{S}, \mathcal{A}, \mathcal{R}, C, \mathcal{P}, \gamma \rangle$[2]: instead of observing $r_t \in \mathcal{R}$ at each time $t$ directly (following his action), our RL agent only observes a perturbed version of $r_t$, denoting as $\tilde{r}_t \in \tilde{\mathcal{R}}$. For most of our presentations, we focus on the cases where $\mathcal{R}, \tilde{\mathcal{R}}$ are finite sets; but our results generalize to the continuous reward settings.

The generation of $\tilde{r}$ follows a certain function $C : \mathcal{S} \times \mathcal{R} \to \tilde{\mathcal{R}}$. To let our presentation stay focused, we consider the following simple state-independent[3] flipping error rates model: if the rewards are binary (consider $r_+$ and $r_-$), $\tilde{r}(s_t, a_t, s_{t+1})$ ($\tilde{r}_t$) can be characterized by the following noise rate parameters $e_+, e_-$: $e_+ = \mathbb{P}(\tilde{r}(s_t, a_t, s_{t+1}) = r_-|r(s_t, a_t, s_{t+1}) = r_+)$, $e_- = \mathbb{P}(\tilde{r}(s_t, a_t, s_{t+1}) = r_+|r(s_t, a_t, s_{t+1}) = r_-)$. When the signal levels are beyond binary, suppose there are $M$ outcomes in total, denoting as $[R_0, R_1, \cdots, R_{M-1}]$. $\tilde{r}_t$ will be generated according to the following confusion matrix $\mathbf{C}_{M \times M}$ where each entry $c_{j,k}$ indicates the flipping probability for generating a perturbed outcome: $c_{j,k} = \mathbb{P}(\tilde{r}_t = R_k|r_t = R_j)$. Again we'd like to note that we focus on settings with finite reward levels for most of our paper, but we provide discussions in Section 3.1 on how to handle continuous rewards with discretizations.

In the paper, we do not assume knowing the noise rates (i.e., the reward confusion matrices), which is different from the assumption of knowing them as adopted in many supervised learning works Natarajan et al. (2013). Instead we will estimate the confusion matrices (Section 3.3).

---

[1]$\gamma = 1$ indicates an undiscounted MDP setting (Schwartz, 1993; Sobel, 1994; Kakade, 2003).

[2]The MDP with perturbed reward can equivalently be defined as a tuple $\tilde{\mathcal{M}} = \langle \mathcal{S}, \mathcal{A}, \mathcal{R}, \tilde{\mathcal{R}}, \mathcal{P}, \gamma \rangle$, with the perturbation function $C$ implicitly defined as the difference between $\mathcal{R}$ and $\tilde{\mathcal{R}}$.

[3]The case of state-dependent perturbed reward is discussed in Appendix C.3

# 3 LEARNING WITH PERTURBED REWARDS

In this section, we first introduce an unbiased estimator for binary rewards in our reinforcement learning setting when the error rates are known. This idea is inspired by Natarajan et al. (2013), but we will extend the method to the multi-outcome, as well as the continuous reward settings.

## 3.1 UNBIASED ESTIMATOR FOR TRUE REWARD

With the knowledge of noise rates (reward confusion matrices), we are able to establish an unbiased approximation of the true reward in a similar way as done in Natarajan et al. (2013). We will call such a constructed unbiased reward as a surrogate reward. To give an intuition, we start with replicating the results for binary reward $\mathcal{R} = \{r_-, r_+\}$ in our RL setting:

**Lemma 1.** *Let $r$ be bounded. Then, if we define,*

$$\hat{r}(s_t, a_t, s_{t+1}) := \begin{cases} \frac{(1-e_-)\cdot r_+ - e_+\cdot r_-}{1-e_+ - e_-} & (\tilde{r}(s_t, a_t, s_{t+1}) = r_+) \\ \frac{(1-e_+)\cdot r_- - e_-\cdot r_+}{1-e_+ - e_-} & (\tilde{r}(s_t, a_t, s_{t+1}) = r_-) \end{cases} \tag{1}$$

*we have for any $r(s_t, a_t, s_{t+1})$, $\mathbb{E}_{\tilde{r}|r}[\hat{r}(s_t, a_t, s_{t+1})] = r(s_t, a_t, s_{t+1})$.*

In the standard supervised learning setting, the above property guarantees convergence - as more training data are collected, the empirical surrogate risk converges to its expectation, which is the same as the expectation of the true risk (due to unbiased estimators). This is also the intuition why we would like to replace the reward terms with surrogate rewards in our RL algorithms.

The above idea can be generalized to the multi-outcome setting in a fairly straight-forward way. Define $\hat{\mathbf{R}} := [\hat{r}(\tilde{r} = R_0), \hat{r}(\tilde{r} = R_1), ..., \hat{r}(\tilde{r} = R_{M-1})]$, where $\hat{r}(\tilde{r} = R_m)$ denotes the value of the surrogate reward when the observed reward is $R_k$. Let $\mathbf{R} = [R_0; R_1; \cdots; R_{M-1}]$ be the bounded reward matrix with $M$ values. We have the following results:

**Lemma 2.** *Suppose $\mathbf{C}_{M \times M}$ is invertible. With defining:*

$$\hat{\mathbf{R}} = \mathbf{C}^{-1} \cdot \mathbf{R}. \tag{2}$$

*we have for any $r(s_t, a_t, s_{t+1})$, $\mathbb{E}_{\tilde{r}|r}[\hat{r}(s_t, a_t, s_{t+1})] = r(s_t, a_t, s_{t+1})$.*

**Continuous reward** When the reward signal is continuous, we discretize it into $M$ intervals and view each interval as a reward level, with its value approximated by its middle point. With increasing $M$, this quantization error can be made arbitrarily small. Our method is then the same as the solution for the multi-outcome setting, except for replacing rewards with discretized ones. Note that the finer-degree quantization we take, the smaller the quantization error - but we would suffer from learning a bigger reward confusion matrix. This is a trade-off question that can be addressed empirically.

So far we have assumed knowing the confusion matrices, but we will address this additional estimation issue in Section 3.3, and present our complete algorithm therein.

## 3.2 CONVERGENCE AND SAMPLE COMPLEXITY: $Q$-LEARNING

We now analyze the convergence and sample complexity of our surrogate reward based RL algorithms (with assuming knowing $\mathbf{C}$), taking $Q$-Learning as an example.

**Convergence guarantee** First, the convergence guarantee is stated in the following theorem:

**Theorem 1.** *Given a finite MDP, denoting as $\hat{\mathcal{M}} = \langle \mathcal{S}, \mathcal{A}, \hat{\mathcal{R}}, \mathcal{P}, \gamma \rangle$, the Q-learning algorithm with surrogate rewards, given by the update rule,*

$$Q_{t+1}(s_t, a_t) = (1 - \alpha_t)Q(s_t, a_t) + \alpha_t \left[ \hat{r}_t + \gamma \max_{b \in \mathcal{A}} Q(s_{t+1}, b) \right], \tag{3}$$

*converges w.p.1 to the optimal Q-function as long as $\sum_t \alpha_t = \infty$ and $\sum_t \alpha_t^2 < \infty$.*

Note that the term on the right hand of Eqn. (3) includes surrogate reward $\hat{r}$ estimated using Eqn. (1) and Eqn. (2). Theorem 1 states that that agents will converge to the optimal policy *w.p.1* with replacing the rewards with surrogate rewards, despite of the noises in observing rewards. This result is not surprising - though the surrogate rewards introduce larger variance, we are grateful of their unbiasedness, which grants us the convergence. In other words, the addition of the perturbed reward does not destroy the convergence guarantees of $Q$-Learning.

**Sample complexity** To establish our sample complexity results, we first introduce a *generative model* following previous literature (Kearns & Singh, 1998; 2000; Kearns et al., 1999). This is a practical MDP setting to simplify the analysis.

**Definition 1.** *A generative model $G(\mathcal{M})$ for an MDP $\mathcal{M}$ is a sampling model which takes a state-action pair $(s_t, a_t)$ as input, and outputs the corresponding reward $r(s_t, a_t)$ and the next state $s_{t+1}$ randomly with the probability of $\mathbb{P}_a(s_t, s_{t+1})$, i.e., $s_{t+1} \sim \mathbb{P}(\cdot|s, a)$.*

Exact value iteration is impractical if the agents follow the generative models above exactly (Kakade, 2003). Consequently, we introduce a *phased Q-Learning* which is similar to the ones presented in Kakade (2003); Kearns & Singh (1998) for the convenience of proving our sample complexity results. We briefly outline *phased Q-Learning* as follows - the complete description (Algorithm 2) can be found in Appendix A.

**Definition 2.** *Phased Q-Learning algorithm takes $m$ samples per phase by calling generative model $G(\mathcal{M})$. It uses the collected $m$ samples to estimate the transition probability $\mathcal{P}$ and update the estimated value function per phase. Calling generative model $G(\hat{\mathcal{M}})$ means that surrogate rewards are returned and used to update value function per phase.*

The sample complexity of *Phased Q-Learning* is given as follows:

**Theorem 2.** *(Upper Bound) Let $r \in [0, R_{\max}]$ be bounded reward, $\mathbf{C}$ be an invertible reward confusion matrix with $\det(\mathbf{C})$ denoting its determinant. For an appropriate choice of $m$, the Phased Q-Learning algorithm calls the generative model $G(\hat{\mathcal{M}})$ $O\left(\frac{|\mathcal{S}||\mathcal{A}|T}{\epsilon^2(1-\gamma)^2\det(\mathbf{C})^2} \log \frac{|\mathcal{S}||\mathcal{A}|T}{\delta}\right)$ times in $T$ epochs, and returns a policy such that for all state $s \in \mathcal{S}$, $|V_\pi(s) - V^*(s)| \leq \epsilon$, $\epsilon > 0$, w.p. $\geq 1 - \delta$, $0 < \delta < 1$.*

Theorem 2 states that, to guarantee the convergence to the optimal policy, the number of samples needed is no more than $O(1/\det(\mathbf{C})^2)$ times of the one needed when the RL agent observes true rewards perfectly. This additional constant is the price we pay for the noise presented in our learning environment. When the noise level is high, we expect to see a much higher $1/\det(\mathbf{C})^2$; otherwise when we are in a low-noise regime , $Q$-Learning can be very efficient with surrogate reward (Kearns & Singh, 2000). Note that Theorem 2 gives the upper bound in discounted MDP setting; for undiscounted setting ($\gamma = 1$), the upper bound is at the order of $O\left(\frac{|\mathcal{S}||\mathcal{A}|T^3}{\epsilon^2\det(\mathbf{C})^2} \log \frac{|\mathcal{S}||\mathcal{A}|T}{\delta}\right)$. Lower bound result is omitted due to the lack of space. The idea of constructing MDP in which learning is difficult and the algorithm must make $\left(\frac{|\mathcal{S}||\mathcal{A}|T}{\epsilon} \log \frac{1}{\delta}\right)$ calls to $G(\hat{\mathcal{M}})$, is similar to Kakade (2003).

While the surrogate reward guarantees the unbiasedness, we sacrifice the variance at each of our learning steps, and this in turn delays the convergence (as also evidenced in the sample complexity bound). It can be verified that the variance of surrogate reward is bounded when $\mathbf{C}$ is invertible, and it is always higher than the variance of true reward. This is summarized in the following theorem:

**Theorem 3.** *Let $r \in [0, R_{\max}]$ be bounded reward and confusion matrix $\mathbf{C}$ is invertible. Then, the variance of surrogate reward $\hat{r}$ is bounded as follows:* $\mathbf{Var}(r) \leq \mathbf{Var}(\hat{r}) \leq \frac{M^2}{\det(\mathbf{C})^2} \cdot R_{\max}^2$.

To give an intuition of the bound, when we have binary reward, the variance for surrogate reward bounds as follows: $\mathbf{Var}(r) \leq \mathbf{Var}(\hat{r}) \leq \frac{4R_{\max}^2}{(1-e_+-e_-)^2}$. As $e_- + e_+ \to 1$, the variance becomes unbounded and the proposed estimator is no longer effective, nor will it be well-defined. In practice, there is a trade-off question between bias and variance by tuning a linear combination of $\mathbf{R}$ and $\hat{\mathbf{R}}$, i.e., $\mathbf{R}_{proxy} = \eta\mathbf{R} + (1-\eta)\hat{\mathbf{R}}$, and choosing an appropriate $\eta \in [0, 1]$.

## 3.3 ESTIMATION OF CONFUSION MATRICES

In Section 3.1 we have assumed the knowledge of reward confusion matrices, in order to compute the surrogate reward. This knowledge is often not available in practice. Estimating these confusion matrices is challenging without knowing any ground truth reward information; but we'd like to note that efficient algorithms have been developed to estimate the confusion matrices in supervised learning settings (Bekker & Goldberger, 2016; Liu & Liu, 2017; Khetan et al., 2017; Hendrycks et al., 2018). The idea in these algorithms is to dynamically refine the error rates based on aggregated rewards. Note this approach is not different from the inference methods in aggregating crowdsourcing

labels, as referred in the literature (Dawid & Skene, 1979; Karger et al., 2011; Liu et al., 2012). We adapt this idea to our reinforcement learning setting, which is detailed as follows.

At each training step, the RL agent collects the noisy reward and the current *state-action* pair. Then, for each pair in $\mathcal{S} \times \mathcal{A}$, the agent predicts the true reward based on accumulated historical observations of reward for the corresponding *state-action* pair via, e.g., averaging (majority voting). Finally, with the predicted true reward and the accuracy (error rate) for each state-action pair, the estimated reward confusion matrices $\tilde{\mathbf{C}}$ are given by

$$\tilde{c}_{i,j} = \frac{\sum_{(s,a)\in\mathcal{S}\times\mathcal{A}} \#\left[\tilde{r}(s,a) = R_j | \bar{r}(s,a) = R_i\right]}{\sum_{(s,a)\in\mathcal{S}\times\mathcal{A}} \#\left[\bar{r}(s,a) = R_i\right]}, \tag{4}$$

where in above $\#[\cdot]$ denotes the number of state-action pair that satisfies the condition $[\cdot]$ in the set of observed rewards $\tilde{R}(s,a)$ (see Algorithm 1 and 3); $\bar{r}(s,a)$ and $\tilde{r}(s,a)$ denote predicted true rewards (using majority voting) and observed rewards when the state-action pair is $(s,a)$. The above procedure of updating $\tilde{c}_{i,j}$ continues indefinitely as more observation arrives.

Our final definition of surrogate reward replaces a known reward confusion $\mathbf{C}$ in Eqn. (2) with our estimated one $\tilde{\mathbf{C}}$. We denote this estimated surrogate reward as $\dot{r}$.

We present (*Reward Robust RL*) in Algorithm 1[4]. Note that the algorithm is rather generic, and we can plug in any exisitng RL algorithm into our reward robust one, with only changes in re-placing the rewards with our es-timated surrogate rewards.

---

**Algorithm 1** Reward Robust RL (sketch)

---

**Input:** $\tilde{\mathcal{M}}, \alpha, \beta, \tilde{R}(s,a)$
**Output:** $Q(s), \pi(s,t)$
    Initialize value function $Q(s,a)$ arbitrarily.
    **while** $Q$ is not converged **do**
        Initialize state $s \in \mathcal{S}$
        **while** $s$ is not terminal **do**
            Choose $a$ from $s$ using policy derived from $Q$
            Take action $a$, observe $s'$ and noisy reward $\tilde{r}$
            **if** collecting enough $\tilde{r}$ for every $\mathcal{S} \times \mathcal{A}$ pair **then**
                Get predicted true reward $\bar{r}$ using majority voting
                Estimate confusion matrix $\tilde{\mathbf{C}}$ based on $\tilde{r}$ and $\bar{r}$ (Eqn. 4)
            Obtain surrogate reward $\dot{r}$ ($\hat{\mathbf{R}} = (1-\eta)\cdot\mathbf{R} + \eta\cdot\mathbf{C}^{-1}\mathbf{R}$)
            Update $Q$ using surrogate reward
            $s \leftarrow s'$
    **return** $Q(s)$ and $\pi(s)$

---

## 4 EXPERIMENTS

In this section, reward robust RL is tested in different games, with different noise settings. Due to space limit, more experimental results can be found in Appendix D.

### 4.1 EXPERIMENTAL SETUP

**Environments and RL Algorithms**  To fully test the performance under different environments, we evaluate the proposed robust reward RL method on two classic control games (CartPole, Pendu-lum) and seven Atari 2600 games (AirRaid, Alien, Carnival, MsPacman, Pong, Phoenix, Seaquest), which encompass a large variety of environments, as well as rewards. Specifically, the rewards could be unary (CartPole), binary (most of Atari games), multivariate (Pong) and even continu-ous (Pendulum). A set of state-of-the-art reinforcement learning algorithms are experimented with while training under different amounts of noise (See Table 3)[5]. For each game and algorithm, three policies are trained based on different random initialization to decrease the variance.

**Reward Post-Processing**  For each game and RL algorithm, we test the performances for learning with true rewards, learning with noisy rewards and learning with surrogate rewards. Both symmet-ric and asymmetric noise settings with different noise levels are tested. For symmetric noise, the confusion matrices are symmetric. As for asymmetric noise, two types of random noise are tested: 1) *rand-one*, each reward level can only be perturbed into another reward; 2) *rand-all*, each reward could be perturbed to any other reward, via adding a random noise matrix. To measure the amount of noise *w.r.t* confusion matrices, we define the weight of noise $\omega$ in Appendix B.2. The larger $\omega$ is, the higher the noise rates are.

---

[4]One complete $Q$-Learning implementation (Algorithm 3) is provided in Appendix C.1.
[5]The detailed settings are accessible in Appendix B.

## 4.2 ROBUSTNESS EVALUATION

**CartPole** The goal in *CartPole* is to prevent the pole from falling by controlling the cart's direction and velocity. The reward is $+1$ for every step taken, including the termination step. When the cart or pole deviates too much or the episode length is longer than 200, the episode terminates. Due to the unary reward $\{+1\}$ in CartPole, a corrupted reward $-1$ is added as the unexpected error ($e_- = 0$). As a result, the reward space $\mathcal{R}$ is extended to $\{+1, -1\}$. Five algorithms $Q$-Learning (1992), CEM (2006), SARSA (1998), DQN (2016) and DDQN (2016) are evaluated.

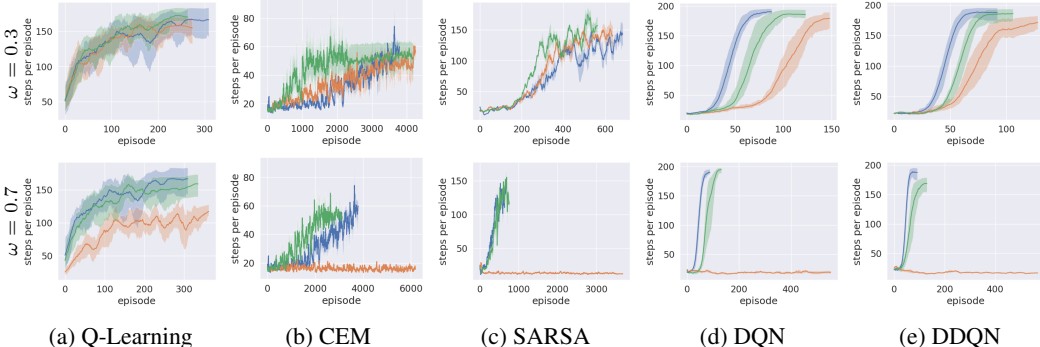

(a) Q-Learning  (b) CEM  (c) SARSA  (d) DQN  (e) DDQN

Figure 1: Learning curves from five RL algorithms on CartPole game with true rewards ($r$) ■, noisy rewards ($\tilde{r}$) ■ and estimated surrogate rewards ($\dot{r}$) ($\eta = 1$) ■. Note that reward confusion matrices C are unknown to the agents here. Full results are in Appendix D.2 (Figure 6).

In Figure 1, we show that our estimator successfully produces meaningful surrogate rewards that adapt the underlying RL algorithms to the noisy settings, without any assumption of the true distribution of rewards. With the noise rate increasing (from 0.1 to 0.9), the models with noisy rewards converge slower due to larger biases. However, we observe that the models always converge to the best score 200 with the help of surrogate rewards.

In some circumstances (slight noise - see Figure 6a, 6b, 6c, 6d), the surrogate rewards even lead to faster convergence. This points out an interesting observation: learning with surrogate reward even outperforms the case with observing the true reward. We conjecture that the way of adding noise and then removing the bias introduces implicit exploration. This implies that for settings even with true reward, we might consider manually adding noise and then remove it in expectation.

**Pendulum** The goal in *Pendulum* is to keep a frictionless pendulum standing up. Different from the CartPole setting, the rewards in pendulum are continuous: $r \in (-16.28, 0.0]$. The closer the reward is to zero, the better performance the model achieves. Following our extension (see Section 3.1), the $(-17, 0]$ is firstly discretized into 17 intervals: $(-17, -16], (-16, -15], \cdots, (-1, 0]$, with its value approximated using its maximum point. After the quantization step, the surrogate rewards can be estimated using multi-outcome extensions presented in Section 3.1.

Table 1: Average scores of various RL algorithms on CartPole and Pendulum with noisy rewards ($\tilde{r}$) and surrogate rewards under known ($\hat{r}$) or estimated ($\dot{r}$) noise rates. Note that the results for last two algorithms DDPG (rand-one) & NAF (rand-all) are on Pendulum, but the others are on CartPole.

| Noise Rate | Reward | Q-Learn | CEM | SARSA | DQN | DDQN | DDPG | NAF |
|---|---|---|---|---|---|---|---|---|
| $\omega = 0.1$ | $\tilde{r}$ | 170.0 | 98.1 | 165.2 | 187.2 | **187.8** | -1.03 | -4.48 |
| | $\hat{r}$ | 165.8 | **108.9** | **173.6** | **200.0** | 181.4 | **-0.87** | **-0.89** |
| | $\dot{r}$ | **181.9** | 99.3 | 171.5 | **200.0** | 185.6 | -0.90 | -1.13 |
| $\omega = 0.3$ | $\tilde{r}$ | 134.9 | 28.8 | 144.4 | 173.4 | 168.6 | -1.23 | -4.52 |
| | $\hat{r}$ | 149.3 | **85.9** | 152.4 | 175.3 | 198.7 | **-1.03** | **-1.15** |
| | $\dot{r}$ | **161.1** | 82.2 | **159.6** | **186.7** | **200.0** | -1.05 | -1.36 |

We experiment two popular algorithms, DDPG (2015) and NAF (2016) in this game. In Figure 2, both algorithms perform well with surrogate rewards under different amounts of noise. In most cases, the biases were corrected in the long-term, even when the amount of noise is extensive (e.g., $\omega = 0.7$). The quantitative scores on CartPole and Pendulum are given in Table 1, where the

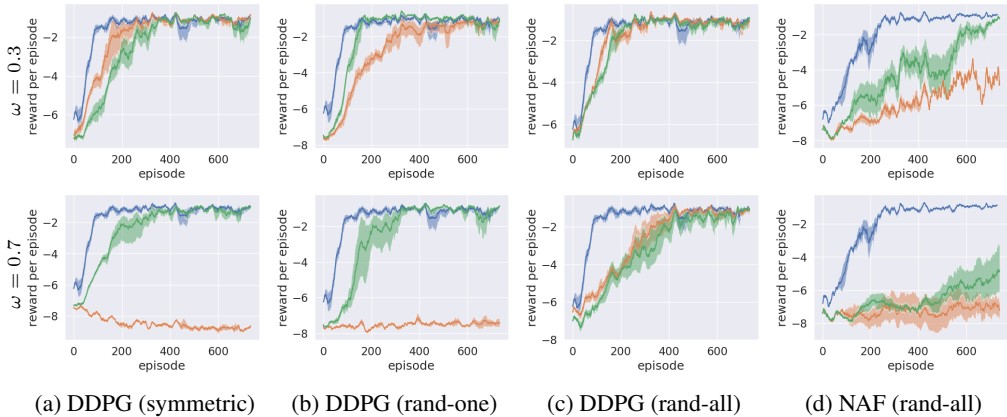

Figure 2: Learning curves from DDPG and NAF on Pendulum game with true rewards ($r$) ■, noisy rewards ($\tilde{r}$) ■ and surrogate rewards ($\hat{r}$) ($\eta = 1$) ■. Both symmetric and asymmetric noise are conduced in the experiments. Full results are in Appendix D.2 (Figure 8).

scores are averaged based on the last thirty episodes. The full results ($\omega > 0.5$) can be found in Appendix D.1, so does Table 2. Our reward robust RL method is able to achieve consistently good scores.

**Atari**  We validate our algorithm on seven Atari 2600 games using the state-of-the-art algorithm PPO (Schulman et al., 2017). The games are chosen to cover a variety of environments. The rewards in the Atari games are clipped into $\{-1, 0, 1\}$. We leave the detailed settings to Appendix B.

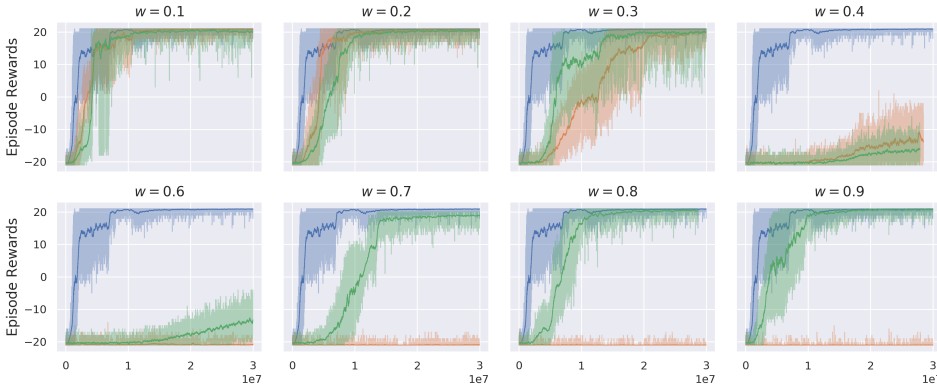

Figure 3: Learning curves from PPO on Pong-v4 game with true rewards ($r$) ■, noisy rewards ($\tilde{r}$) ■ and surrogate rewards ($\eta = 1$) ($\hat{r}$) ■. The noise rates increase from 0.1 to 0.9, with a step of 0.1.

Results for PPO on Pong-v4 in symmetric noise setting are presented in Figure 3. Due to limited space, more results on other Atari games and noise settings are given in Appendix D.3. Similar to previous results, our surrogate estimator performs consistently well and helps PPO converge to the optimal policy. Table 2 shows the average scores of PPO on five selected Atari games with different amounts of noise (symmetric & asymmetric). In particular, when the noise rates $e_+ = e_- > 0.3$, agents with surrogate rewards obtain significant amounts of improvements in average scores. We do not present the results for the case with unknown $\mathbf{C}$ because the state-space (image-input) is very large for Atari games, which is difficult to handle with the solution given in Section 3.3.

## 5 CONCLUSION

Only an underwhelming amount of reinforcement learning studies have focused on the settings with perturbed and noisy rewards, despite the fact that such noises are common when exploring a real-world scenario, that faces sensor errors or adversarial examples. We adapt the ideas from supervised

Table 2: Average scores of PPO on five selected games with noisy rewards ($\tilde{r}$) and surrogate rewards ($\hat{r}$). The experiments are repeated three times with different random seeds.

| Noise Rate | Reward | Lift (↑) | Mean | Alien | Carnival | Phoenix | MsPacman | Seaquest |
|---|---|---|---|---|---|---|---|---|
| $e_- = e_+ = 0.1$ | $\tilde{r}$ | - | 2044.2 | **1814.8** | 1239.2 | 4608.9 | 1709.1 | 849.2 |
| | $\hat{r}$ | **67.5%**↑ | **3423.1** | 1741.0 | **3630.3** | **7586.3** | **2547.3** | **1610.6** |
| $e_- = 0.1, e_+ = 0.3$ | $\tilde{r}$ | - | 770.5 | 893.3 | 841.8 | 250.7 | 1151.1 | 715.7 |
| | $\hat{r}$ | **20.3%**↑ | **926.6** | **973.7** | **955.2** | **643.9** | **1307.1** | **753.1** |
| $e_- = e_+ = 0.3$ | $\tilde{r}$ | - | 1180.1 | 543.1 | 919.8 | 2600.3 | 1109.6 | **727.8** |
| | $\hat{r}$ | **46.7%**↑ | **1730.8** | **1637.7** | **966.1** | **4171.5** | **1470.2** | 408.6 |

learning with noisy examples (Natarajan et al., 2013), and propose a simple but effective RL framework for dealing with noisy rewards. The convergence guarantee and finite sample complexity of $Q$-Learning (or its variant) with estimated surrogate rewards are given. To validate the effectiveness of our approach, extensive experiments are conducted on OpenAI Gym, showing that surrogate rewards successfully rescue models from misleading rewards even at high noise rates.

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

# A    PROOFS

*Proof of Lemma 1.* For simplicity, we shorthand $\hat{r}(s_t, a_t, s_{t+1}), \tilde{r}(s_t, a_t, s_{t+1}), r(s_t, a_t, s_{t+1})$ as $\hat{r}, \tilde{r}, r$, and let $r_+, r_-, \hat{r}_+, \hat{r}_-$ denote the general reward levels and corresponding surrogate ones:

$$\mathbb{E}_{\tilde{r}|r}(\hat{r}) = \mathbb{P}_{\tilde{r}|r}(\hat{r} = \hat{r}_-)\hat{r}_- + \mathbb{P}_{\tilde{r}|r}(\hat{r} = \hat{r}_+)\hat{r}_+. \tag{5}$$

When $r = r_+$, from the definition in Lemma 1:

$$\mathbb{P}_{\tilde{r}|r}(\hat{r} = \hat{r}_-) = e_+, \ \mathbb{P}_{\tilde{r}|r}(\hat{r} = \hat{r}_+) = 1 - e_+.$$

Taking the definition of surrogate rewards Eqn. (1) into Eqn. (5), we have

$$\mathbb{E}_{\tilde{r}|r}(\hat{r}) = e_+ \cdot \hat{r}_- + (1 - e_+) \cdot \hat{r}_+$$
$$= e_+ \cdot \frac{(1 - e_+)r_- - e_- r_+}{1 - e_- - e_+} + (1 - e_+) \cdot \frac{(1 - e_-)r_+ - e_+ r_-}{1 - e_- - e_+} = r_+.$$

Similarly, when $r = r_-$, it also verifies $\mathbb{E}_{\tilde{r}|r}[\hat{r}(s_t, a_t, s_{t+1})] = r(s_t, a_t, s_{t+1})$.    □

*Proof of Lemma 2.* The idea of constructing unbiased estimator is easily adapted to multi-outcome reward settings via writing out the conditions for the unbiasedness property (s.t. $\mathbb{E}_{\tilde{r}|r}[\hat{r}] = r$.). For simplicity, we shorthand $\hat{r}(\tilde{r} = R_i)$ as $\hat{R}_i$ in the following proofs. Similar to Lemma 1, we need to solve the following set of functions to obtain $\hat{r}$:

$$\begin{cases} R_0 = c_{0,0} \cdot \hat{R}_0 + c_{0,1} \cdot \hat{R}_1 + \cdots + c_{0,M-1} \cdot \hat{R}_{M-1} \\ R_1 = c_{1,0} \cdot \hat{R}_0 + c_{1,1} \cdot \hat{R}_1 + \cdots + c_{1,M-1} \cdot \hat{R}_{M-1} \\ \cdots \\ R_{M-1} = c_{M-1,0} \cdot \hat{R}_0 + c_{M-1,1} \cdot \hat{R}_1 + \cdots + c_{M-1,M-1} \cdot \hat{R}_{M-1} \end{cases}$$

where $\hat{R}_i$ denotes the value of the surrogate reward when the observed reward is $R_i$. Define $\mathbf{R} := [R_0; R_1; \cdots ; R_{M-1}]$, and $\hat{\mathbf{R}} := [\hat{R}_0, \hat{R}_1, ..., \hat{R}_{M-1}]$, then the above equations are equivalent to: $\mathbf{R} = \mathbf{C} \cdot \hat{\mathbf{R}}$. If the confusion matrix $\mathbf{C}$ is invertible, we obtain the surrogate reward:

$$\hat{\mathbf{R}} = \mathbf{C}^{-1} \cdot \mathbf{R}.$$

According to above definition, for any true reward level $R_i, i = 0, 1, \cdots , M - 1$, we have

$$\mathbb{E}_{\tilde{r}|r=R_i}[\hat{r}] = c_{i,0} \cdot \hat{R}_0 + c_{i,1} \cdot \hat{R}_1 + \cdots + c_{i,M-1} \cdot \hat{R}_{M-1} = R_i.$$

□

Furthermore, the probabilities for observing surrogate rewards can be written as follows:

$$\hat{\mathbf{P}} = [\hat{p}_1, \hat{p}_2, \cdots , \hat{p}_M] = \left[ \sum_j p_j c_{j,1}, \sum_j p_j c_{j,2}, \cdots , \sum_j p_j c_{j,M} \right],$$

where $\hat{p}_i = \sum_j p_j c_{j,i}$, and $\hat{p}_i$, $p_i$ represent the probabilities of occurrence for surrogate reward $\hat{R}_i$ and true reward $R_i$ respectively.

**Corollary 1.** *Let $\hat{p}_i$ and $p_i$ denote the probabilities of occurrence for surrogate reward $\hat{r}(\tilde{r} = R_i)$ and true reward $R_i$. Then the surrogate reward satisfies,*

$$\sum_{s' \in \mathcal{S}} \mathbb{P}_a(s_t, s_{t+1}) r(s_t, a, s_{t+1}) = \sum_j p_j R_j = \sum_j \hat{p}_j \hat{R}_j. \tag{6}$$

*Proof of Corollary 1.* From Lemma 2, we have,

$$\sum_{s_t \in \mathcal{S}} \mathbb{P}_a(s_t, s_{t+1}) r(s_t, a, s_{t+1}) = \sum_{s_{t+1} \in \mathcal{S}; R_j \in \mathcal{R}} \mathbb{P}_a(s_t, s_{t+1}, R_j) R_j$$
$$= \sum_{R_j \in \mathcal{R}} \sum_{s_{t+1} \in \mathcal{S}} \mathbb{P}_a(s_t, s_{t+1}) R_j = \sum_{R_j \in \mathcal{R}} p_j R_j = \sum_j p_j R_j.$$

Consequently,

$$\sum_j \hat{p}_j \hat{R}_j = \sum_j \sum_k p_k c_{k,j} \hat{R}_j = \sum_k p_k \sum_j c_{k,j} \hat{R}_j$$

$$= \sum_k p_k R_k = \sum_{s_t \in \mathcal{S}} \mathbb{P}_a(s_t, s_{t+1}) r(s_t, a, s_{t+1}).$$

$\square$

To establish Theorem 1, we need an auxiliary result (Lemma 3) from stochastic process approximation, which is widely adopted for the convergence proof for $Q$-Learning (Jaakkola et al., 1993; Tsitsiklis, 1994).

**Lemma 3.** *The random process $\{\Delta_t\}$ taking values in $\mathbb{R}^n$ and defined as*

$$\Delta_{t+1}(x) = (1 - \alpha_t(x))\Delta_t(x) + \alpha_t(x)F_t(x)$$

*converges to zero w.p.1 under the following assumptions:*

- $0 \leq \alpha_t \leq 1$, $\sum_t \alpha_t(x) = \infty$ and $\sum_t \alpha_t(x)^2 < \infty$;

- $||\mathbb{E}\left[F_t(x)|\mathcal{F}_t\right]||_W \leq \gamma ||\Delta_t||$, *with* $\gamma < 1$;

- *var* $\left[F_t(x)|\mathcal{F}_t\right] \leq C(1 + ||\Delta_t||_W^2)$, *for* $C > 0$.

*Here $\mathcal{F}_t = \{\Delta_t, \Delta_{t-1}, \cdots, F_{t-1} \cdots, \alpha_t, \cdots\}$ stands for the past at step $t$, $\alpha_t(x)$ is allowed to depend on the past insofar as the above conditions remain valid. The notation $||\cdot||_W$ refers to some weighted maximum norm.*

*Proof of Lemma 3.* See previous literature (Jaakkola et al., 1993; Tsitsiklis, 1994). $\square$

*Proof of Theorem 1.* For simplicity, we abbreviate $s_t$, $s_{t+1}$, $Q_t$, $Q_{t+1}$, $r_t$, $\hat{r}_t$ and $\alpha_t$ as $s$, $s'$, $Q$, $Q'$, $r$, $\hat{r}$, and $\alpha$, respectively.

Subtracting from both sides the quantity $Q^*(s, a)$ in Eqn. (3):

$$Q'(s, a) - Q^*(s, a) = (1 - \alpha)\left(Q(s, a) - Q^*(s, a)\right) + \alpha \left[\hat{r} + \gamma \max_{b \in \mathcal{A}} Q(s', b) - Q^*(s, a)\right].$$

Let $\Delta_t(s, a) = Q(s, a) - Q^*(s, a)$ and $F_t(s, a) = \hat{r} + \gamma \max_{b \in \mathcal{A}} Q(s', b) - Q^*(s, a)$.

$$\Delta_{t+1}(s', a) = (1 - \alpha)\Delta_t(s, a) + \alpha F_t(s, a).$$

In consequence,

$$\mathbb{E}\left[F_t(x)|\mathcal{F}_t\right] = \sum_{s' \in \mathcal{S}; \hat{r} \in \mathcal{R}} \mathbb{P}_a(s, s', \hat{r}) \left[\hat{r} + \gamma \max_{b \in \mathcal{A}} Q(s', b)\right] - Q^*(s, a)$$

$$= \sum_{s' \in \mathcal{S}; \hat{r} \in \mathcal{R}} \mathbb{P}_a(s, s', \hat{r})\hat{r} + \sum_{s' \in \mathcal{S}} \mathbb{P}_a(s, s') \left[\gamma \max_{b \in \mathcal{A}} Q(s', b) - r - \gamma \max_{b \in \mathcal{A}} Q^*(s', b)\right]$$

$$= \sum_{s' \in \mathcal{S}; \hat{r} \in \mathcal{R}} \mathbb{P}_a(s, s', \hat{r})\hat{r} - \sum_{s' \in \mathcal{S}} \mathbb{P}_a(s, s')r + \sum_{s' \in \mathcal{S}} \mathbb{P}_a(s, s')\gamma \left[\max_{b \in \mathcal{A}} Q(s', b) - \max_{b \in \mathcal{A}} Q^*(s', b)\right]$$

$$= \sum_j \hat{p}_j \hat{r}_j - \sum_{s' \in \mathcal{S}} \mathbb{P}_a(s, s')r + \sum_{s' \in \mathcal{S}} \mathbb{P}_a(s, s')\gamma \left[\max_{b \in \mathcal{A}} Q(s', b) - \max_{b \in \mathcal{A}} Q^*(s', b)\right]$$

$$= \sum_{s' \in \mathcal{S}} \mathbb{P}_a(s, s')\gamma \left[\max_{b \in \mathcal{A}} Q(s', b) - \max_{b \in \mathcal{A}} Q^*(s', b)\right] \qquad \text{(using Eqn. (6))}$$

$$\leq \gamma \sum_{s' \in \mathcal{S}} \mathbb{P}_a(s, s') \max_{b \in \mathcal{A}, s' \in \mathcal{S}} |Q(s', b) - Q^*(s', b)|$$

$$= \gamma \sum_{s' \in \mathcal{S}} \mathbb{P}_a(s, s')||Q - Q^*||_\infty = \gamma ||Q - Q^*||_\infty = \gamma ||\Delta_t||_\infty.$$

Finally,

$$
\begin{aligned}
\mathbf{Var}\left[F_t(x)|\mathcal{F}_t\right] &= \mathbb{E}\left[\left(\hat{r} + \gamma \max_{b \in \mathcal{A}} Q(s', b) - \sum_{s' \in \mathcal{S}; \hat{r} \in \mathcal{R}} \mathbb{P}'(s, s', \hat{r})\left[\hat{r} + \gamma \max_{b \in \mathcal{A}} Q(s', b)\right]\right)^2\right] \\
&= \mathbf{Var}\left[\hat{r} + \gamma \max_{b \in \mathcal{A}} Q(s', b)|\mathcal{F}_t\right]
\end{aligned}
$$

.

Because $\hat{r}$ is bounded, it can be clearly verified that

$$
\mathbf{Var}\left[F_t(x)|\mathcal{F}_t\right] \le C(1 + ||\Delta_t||_W^2)
$$

for some constant $C$. Then, due to the Lemma 3, $\Delta_t$ converges to zero w.p.1, *i.e.*, $Q'(s, a)$ converges to $Q^*(s, a)$. $\qquad\square$

The procedure of *Phased Q-Learning* is described as Algorithm 2:

---
**Algorithm 2** Phased $Q$-Learning

---
**Input:** $G(\mathcal{M})$: generative model of $\mathcal{M} = (\mathcal{S}, \mathcal{A}, \mathcal{R}, \mathcal{P}, \gamma)$, $T$: number of iterations.
**Output:** $\hat{V}(s)$: value function, $\hat{\pi}(s, t)$: policy function.
  1: Set $\hat{V}_T(s) = 0$
  2: **for** $t = T - 1, \cdots, 0$ **do**
     1. Calling $G(\mathcal{M})$ $m$ times for each state-action pair.

$$
\hat{\mathbb{P}}_a(s_t, s_{t+1}) = \frac{\#[(s_t, a_t) \to s_{t+1}]}{m}
$$

     2. Set

$$
\hat{V}(s) = \max_{a \in \mathcal{A}} \sum_{s_{t+1} \in \mathcal{S}} \hat{\mathbb{P}}_a(s_t, s_{t+1})\left[r_t + \gamma\hat{V}(s_{t+1})\right]
$$
$$
\hat{\pi}(s, t) = \arg\max_{a \in \mathcal{A}} \hat{V}(s)
$$

  3: **return** $\hat{V}(s)$ and $\hat{\pi}(s, t)$

---

Note that $\hat{\mathbb{P}}$ here is the estimated transition probability, which is different from $\mathbb{P}$ in Eqn. (6).

To obtain the sample complexity results, the range of our surrogate reward needs to be known. Assuming reward $r$ is bounded in $[0, R_{\max}]$, Lemma 4 below states that the surrogate reward is also bounded, when the confusion matrices are invertible:

**Lemma 4.** *Let $r \in [0, R_{\max}]$ be bounded, where $R_{\max}$ is a constant; suppose $\mathbf{C}_{M \times M}$, the confusion matrix, is invertible with its determinant denoting as $\det(\mathbf{C})$. Then the surrogate reward satisfies*

$$
0 \le |\hat{r}| \le \frac{M}{\det(\mathbf{C})} R_{\max}. \tag{7}
$$

*Proof of Lemma 4.* From Eqn. (2), we have,

$$
\hat{\mathbf{R}} = \mathbf{C}^{-1} \cdot \mathbf{R} = \frac{\mathrm{adj}(\mathbf{C})}{\det(\mathbf{C})} \cdot \mathbf{R},
$$

where $\mathrm{adj}(\mathbf{C})$ is the adjugate matrix of $\mathbf{C}$; $\det(\mathbf{C})$ is the determinant of $\mathbf{C}$. It is known from linear algebra that,

$$
\mathrm{adj}(\mathbf{C})_{ij} = (-1)^{i+j} \cdot \mathbf{M}_{ji},
$$

where $\mathbf{M}_{ji}$ is the determinant of the $(M-1) \times (M-1)$ matrix that results from deleting row $j$ and column $i$ of $\mathbf{C}$. Therefore, $\mathbf{M}_{ji}$ is also bounded:

$$\mathbf{M}_{ji} \leq \sum_{\sigma \in S_n} \left( |\mathrm{sgn}(\sigma)| \prod_{m=1}^{M-1} c'_{m,\sigma_n} \right) \leq \prod_{m=0}^{M-1} \left( \sum_{n=0}^{M-1} c_{m,n} \right) = 1^M = 1,$$

where the sum is computed over all permutations $\sigma$ of the set $\{0, 1, \cdots, M-2\}$; $c'$ is the element of $\mathbf{M}_{ji}$; $\mathrm{sgn}(\sigma)$ returns a value that is $+1$ whenever the reordering given by $\sigma$ can be achieved by successively interchanging two entries an even number of times, and $-1$ whenever it can not.

Consequently,

$$\left| \hat{R}_i \right| = \frac{\sum_j |\mathrm{adj}(\mathbf{C})_{ij}| \cdot |R_j|}{\det(\mathbf{C})} \leq \frac{M}{\det(\mathbf{C})} \cdot R_{\max}.$$

$\square$

*Proof of Theorem 2.* From Hoeffding's inequality, we obtain:

$$P \left( \left| \sum_{s_{t+1} \in \mathcal{S}} \mathbb{P}_a(s_t, s_{t+1}) V^*_{t+1}(s_{t+1}) - \sum_{s_{t+1} \in \mathcal{S}} \hat{\mathbb{P}}_a(s_t, s_{t+1}) V^*_{t+1}(s_{t+1}) \right| \geq \epsilon \right)$$
$$\leq 2 \exp \left( \frac{-2m\epsilon^2(1-\gamma)^2}{R^2_{\max}} \right),$$

because $V_t(s_t)$ is bounded within $\frac{R_{\max}}{1-\gamma}$. In the same way, $\hat{r}_t$ is bounded by $\frac{M}{\det(\mathbf{C})} \cdot R_{\max}$ from Lemma 4. We then have,

$$P \left( \left| \sum_{\substack{s_{t+1} \in \mathcal{S} \\ \hat{r}_t \in \hat{\mathcal{R}}}} \mathbb{P}_a(s_t, s_{t+1}, \hat{r}_t) \hat{r}_t - \sum_{\substack{s_{t+1} \in \mathcal{S} \\ \hat{r}_t \in \hat{\mathcal{R}}}} \hat{\mathbb{P}}_a(s_t, s_{t+1}, \hat{r}_t) \hat{r}_t \right| \geq \epsilon \right) \leq 2 \exp \left( \frac{-2m\epsilon^2 \det(\mathbf{C})^2}{M^2 R^2_{\max}} \right).$$

Further, due to the unbiasedness of surrogate rewards, we have

$$\sum_{s_{t+1} \in \mathcal{S}} \mathbb{P}_a(s_t, s_{t+1}) r_t = \sum_{s_{t+1} \in \mathcal{S}; \hat{r}_t \in \hat{\mathcal{R}}} \mathbb{P}_a(s_t, s_{t+1}, \hat{r}_t) \hat{r}_t.$$

As a result,

$$\left| V^*_t(s) - \hat{V}_t(s) \right| = \max_{a \in \mathcal{A}} \sum_{s_{t+1} \in \mathcal{S}} \mathbb{P}_a(s_t, s_{t+1}) \left[ r_t + \gamma V^*_{t+1}(s_{t+1}) \right]$$

$$- \max_{a \in \mathcal{A}} \sum_{s_{t+1} \in \mathcal{S}} \hat{\mathbb{P}}_a(s_t, s_{t+1}) \left[ \hat{r}_t + \gamma V^*_{t+1}(s_{t+1}) \right]$$

$$\leq \epsilon_1 + \gamma \max_{a \in \mathcal{A}} \left| \sum_{s_{t+1} \in \mathcal{S}} \mathbb{P}_a(s_t, s_{t+1}) V^*_{t+1}(s_{t+1}) - \sum_{s_{t+1} \in \mathcal{S}} \hat{\mathbb{P}}_a(s_t, s_{t+1}) V^*_{t+1}(s_{t+1}) \right|$$

$$+ \max_{a \in \mathcal{A}} \left| \sum_{s_{t+1} \in \mathcal{S}} \mathbb{P}_a(s_t, s_{t+1}) r_t - \sum_{s_{t+1} \in \mathcal{S}; \hat{r}_t \in \hat{\mathcal{R}}} \mathbb{P}_a(s_t, s_{t+1}, \hat{r}_t) \hat{r}_t \right|$$

$$\leq \gamma \max_{s \in \mathcal{S}} \left| V^*_{t+1}(s) - \hat{V}_{t+1}(s) \right| + \epsilon_1 + \gamma \epsilon_2$$

In the same way,

$$\left| V_t(s) - \hat{V}_t(s) \right| \leq \gamma \max_{s \in \mathcal{S}} \left| V^*_{t+1}(s) - \hat{V}_{t+1}(s) \right| + \epsilon_1 + \gamma \epsilon_2$$

Recursing the two equations in two directions $(0 \to T)$, we get

$$\max_{s \in \mathcal{S}} \left| V^*(s) - \hat{V}(s) \right| \leq (\epsilon_1 + \gamma \epsilon_2) + \gamma(\epsilon_1 + \gamma \epsilon_2) + \cdots + \gamma^{T-1}(\epsilon_1 + \gamma \epsilon_2)$$

$$= \frac{(\epsilon_1 + \gamma \epsilon_2)(1 - \gamma^T)}{1 - \gamma}$$

$$\max_{s \in \mathcal{S}} \left| V(s) - \hat{V}(s) \right| \leq \frac{(\epsilon_1 + \gamma \epsilon_2)(1 - \gamma^T)}{1 - \gamma}$$

Combining these two inequalities above we have:

$$\max_{s \in \mathcal{S}} |V^*(s) - V(s)| \leq 2 \frac{(\epsilon_1 + \gamma \epsilon_2)(1 - \gamma^T)}{1 - \gamma} \leq 2 \frac{(\epsilon_1 + \gamma \epsilon_2)}{1 - \gamma}.$$

Let $\epsilon_1 = \epsilon_2$, so $\max_{s \in \mathcal{S}} |V^*(s) - V(s)| \leq \epsilon$ as long as

$$\epsilon_1 = \epsilon_2 \leq \frac{(1 - \gamma)\epsilon}{2(1 + \gamma)}.$$

For arbitrarily small $\epsilon$, by choosing $m$ appropriately, there always exists $\epsilon_1 = \epsilon_2 = \frac{(1-\gamma)\epsilon}{2(1+\gamma)}$ such that the policy error is bounded within $\epsilon$. That is to say, the *Phased Q-Learning* algorithm can converge to the near optimal policy within finite steps using our proposed surrogate rewards.

Finally, there are $|\mathcal{S}||\mathcal{A}|T$ transitions under which these conditions must hold, where $|\cdot|$ represent the number of elements in a specific set. Using a union bound, the probability of failure in any condition is smaller than

$$2|\mathcal{S}||\mathcal{A}|T \cdot \exp\left(-m \frac{\epsilon^2(1 - \gamma)^2}{2(1 + \gamma)^2} \cdot \min\{(1 - \gamma)^2, \frac{\det(\mathbf{C})^2}{M^2}\}\right).$$

We set the error rate less than $\delta$, and $m$ should satisfy that

$$m = O\left(\frac{1}{\epsilon^2(1 - \gamma)^2 \det(\mathbf{C})^2} \log \frac{|\mathcal{S}||\mathcal{A}|T}{\delta}\right).$$

In consequence, after $m|\mathcal{S}||\mathcal{A}|T$ calls, which is, $O\left(\frac{|\mathcal{S}||\mathcal{A}|T}{\epsilon^2(1-\gamma)^2 \det(\mathbf{C})^2} \log \frac{|\mathcal{S}||\mathcal{A}|T}{\delta}\right)$, the value function converges to the optimal one for every state $s$, with probability greater than $1 - \delta$. $\qquad\square$

The above bound is for discounted MDP setting with $0 \leq \gamma < 1$. For undiscounted setting $\gamma = 1$, since the total error (for entire trajectory of $T$ time-steps) has to be bounded by $\epsilon$, therefore, the error for each time step has to be bounded by $\frac{\epsilon}{T}$. Repeating our anayslis, we obtain the following upper bound:

$$O\left(\frac{|\mathcal{S}||\mathcal{A}|T^3}{\epsilon^2 \det(\mathbf{C})^2} \log \frac{|\mathcal{S}||\mathcal{A}|T}{\delta}\right).$$

*Proof of Theorem 3.*

$$\mathbf{Var}(\hat{r}) - \mathbf{Var}(r) = \mathbb{E}\left[(\hat{r} - \mathbb{E}[\hat{r}])^2\right] - \mathbb{E}\left[(r - \mathbb{E}[r])^2\right]$$

$$= \mathbb{E}[\hat{r}^2] - \mathbb{E}[\hat{r}]^2 + \mathbb{E}[r^2] - \mathbb{E}[r]^2$$

$$= \sum_j \hat{p}_j \hat{R}_j^2 - \left(\sum_j \hat{p}_j \hat{R}_j\right)^2 - \left[\sum_j p_j R_j^2 - \left(\sum_j p_j R_j\right)^2\right]$$

$$= \sum_j \hat{p}_j \hat{R}_j^2 - \sum_j p_j R_j^2 = \sum_j \sum_i p_i c_{i,j} \hat{R}_j^2 - \sum_j p_j \left(\sum_i c_{j,i} \hat{R}_i\right)^2$$

$$= \sum_j p_j \left(\sum_i c_{j,i} \hat{R}_i^2 - \left(\sum_i c_{j,i} \hat{R}_i\right)^2\right).$$

Using the CauchySchwarz inequality,

$$\sum_i c_{j,i} \hat{R}_i^{\ 2} = \sum_i \sqrt{c_{j,i}}^2 \cdot \sum_i \left( \sqrt{c_{j,i}} \hat{R}_i \right)^2 \geq \left( \sum_i c_{j,i} \hat{R}_i \right)^2.$$

So we get,

$$\mathbf{Var}(\hat{r}) - \mathbf{Var}(r) \geq 0.$$

In addition,

$$\mathbf{Var}(\hat{r}) = \sum_j \hat{p}_j \hat{R}_j^{\ 2} - \left( \sum_j \hat{p}_j \hat{R}_j \right)^2 \leq \sum_j \hat{p}_j \hat{R}_j^{\ 2}$$

$$\leq \sum_j \hat{p}_j \frac{M^2}{\det(\mathbf{C})^2} \cdot R_{\max}^2 = \frac{M^2}{\det(\mathbf{C})^2} \cdot R_{\max}^2.$$

$\square$

## B  EXPERIMENTAL SETUP

We set up our experiments within the popular OpenAI baselines (Dhariwal et al., 2017) and keras-rl (Plappert, 2016) framework. Specifically, we integrate the algorithms and interact with OpenAI Gym (Brockman et al., 2016) environments (Table 3).

### B.1  RL ALGORITHMS

A set of state-of-the-art reinforcement learning algorithms are experimented with while training under different amounts of noise, including $Q$-Learning (Watkins, 1989; Watkins & Dayan, 1992), Cross-Entropy Method (CEM) (Szita & Lörincz, 2006), Deep SARSA (Sutton & Barto, 1998), Deep $Q$-Network (DQN) (Mnih et al., 2013; 2015; van Hasselt et al., 2016), Dueling DQN (DDQN) (Wang et al., 2016), Deep Deterministic Policy Gradient (DDPG) (Lillicrap et al., 2015), Continuous DQN (NAF) (Gu et al., 2016) and Proximal Policy Optimization (PPO) (Schulman et al., 2017) algorithms. For each game and algorithm, three policies are trained based on different random initialization to decrease the variance in experiments.

Table 3: RL algorithms utilized in the robustness evaluation.

| Environment | RL Algorithm |
| --- | --- |
| CartPole | $Q$-Learning (1989) |
| | CEM (2006) |
| | SARSA (1998) |
| | DQN (2013; 2015) |
| | DDQN (2016) |
| Pendulum | DDPG (2015) |
| | NAF (2016) |
| Atari Games | PPO (2017) |

### B.2  POST-PROCESSING REWARDS

We explore both symmetric and asymmetric noise of different noise levels. For symmetric noise, the confusion matrices are symmetric, which means the probabilities of corruption for each reward choice are equivalent. For instance, a confusion matrix

$$\mathbf{C} = \begin{bmatrix} 0.8 & 0.2 \\ 0.2 & 0.8 \end{bmatrix}$$

says that $r_1$ could be corrupted into $r_2$ with a probability of 0.2 and so does $r_2$ (weight = 0.2).

As for asymmetric noise, two types of random noise are tested: 1) *rand-one*, each reward level can only be perturbed into another reward; 2) *rand-all*, each reward could be perturbed to any other reward. To measure the amount of noise *w.r.t* confusion matrices, we define the weight of noise as follows:

$$\mathbf{C} = (1 - \omega) \cdot \mathbf{I} + \omega \cdot \mathbf{N},\ \omega \in [0, 1],$$

where $\omega$ controls the weight of noise; $\mathbf{I}$ and $\mathbf{N}$ denote the identity and noise matrix respectively. Suppose there are $M$ outcomes for true rewards, $\mathbf{N}$ writes as:

$$\mathbf{N} = \begin{bmatrix} n_{0,0} & n_{0,1} & \cdots & n_{0,M-1} \\ \cdots & \cdots & \cdots & \cdots \\ n_{M-1,0} & n_{M-1,1} & \cdots & n_{M-1,M-1} \end{bmatrix},$$

where for each row $i$, 1) rand-one: randomly choose $j$, *s.t* $n_{i,j} = 1$ and $n_{i,k} \neq 0$ if $k \neq j$; 2) rand-all: generate $M$ random numbers that sum to 1, *i.e.*, $\sum_j n_{i,j} = 1$. For the simplicity, for symmetric noise, we choose $\mathbf{N}$ as an anti-identity matrix. As a result, $c_{i,j} = 0$, if $i \neq j$ or $i + j \neq M$.

## B.3 PERTURBED-REWARD MDP EXAMPLE

To obtain an intuitive view of the reward perturbation model, where the observed rewards are generated based on a reward confusion matrix, we constructed a simple MDP and evaluated the performance of *robust reward Q-Learning* (Algorithm 1) on different noise ratios (both symmetric and asymmetric). The finite MDP is formulated as Figure 4a: when the agent reaches state 5, it gets an instant reward of $r_+ = 1$, otherwise a zero reward $r_- = 0$. During the explorations, the rewards are perturbed according to the confusion matrix $\mathbf{C}_{2\times2} = [1 - e_-, e_-; e_+, 1 - e_+]$.

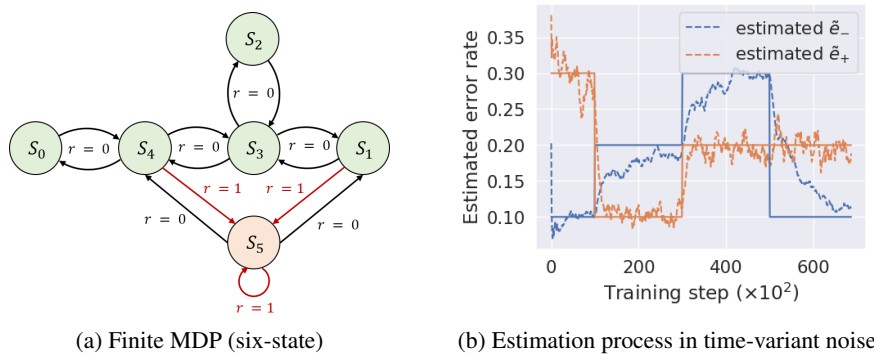

(a) Finite MDP (six-state)      (b) Estimation process in time-variant noise

Figure 4: Perturbed-Reward MDP Example

There are two experiments conducted in this setting: 1) performance of $Q$-Learning under different noise rates (Table 4); 2) robustness of estimation module *in time-variant noise* (Figure 4b). As shown in Table 4, $Q$-Learning achieved better results consistently with the guidance of surrogate rewards and the confusion matrix estimation algorithm. For time-variant noise, we generated varying amount of noise at different training stages: 1) $e_- = 0.1, e_+ = 0.3$ (0 to $1e^4$ steps); 2) $e_- = 0.2, e_+ = 0.1$ ($1e^4$ to $3e^4$ steps); 3) $e_- = 0.3, e_+ = 0.2$ ($3e^4$ to $5e^4$ steps); 4) $e_- = 0.1, e_+ = 0.2$ ($5e^4$ to $7e^4$ steps). In Figure 4b, we show that Algorithm 1 is robust against time-variant noise, which dynamically adjusts the estimated $\tilde{\mathbf{C}}$ after the noise distribution changes. Note that we set a maximum memory size for collected noisy rewards to let the agents only learn with recent observations.

Table 4: Average performance of $Q$-Learning on Perturbed MDP example (Figure 4a) with noisy rewards ($\tilde{r}$), surrogate rewards under known ($\hat{r}$) or estimated ($\dot{r}$) noise rates. Note that the success means the agents can find the optimal policy at every initial state according to learned $Q$-function. We repeated the experiments 1,000 times to calculate the successful rate for each noisy setting.

| Noise Rate | Reward | Lift (↑) | Success | Noise Rate | Reward | Lift (↑) | Success |
|---|---|---|---|---|---|---|---|
| | $\tilde{r}$ | - | 93.4% | | $\tilde{r}$ | - | 89.5% |
| $e_- = e_+ = 0.1$ | $\hat{r}$ | **2.7%** ↑ | 96.1% | $e_- = 0.1, e_+ = 0.3$ | $\hat{r}$ | **3.4%** ↑ | 92.9% |
| | $\dot{r}$ | **3.2%** ↑ | 96.6% | | $\dot{r}$ | **2.9%** ↑ | 92.4% |
| | $\tilde{r}$ | - | 85.5% | | $\tilde{r}$ | - | 90.3% |
| $e_- = e_+ = 0.3$ | $\hat{r}$ | **1.4%** ↑ | 86.9% | $e_- = 0.3, e_+ = 0.1$ | $\hat{r}$ | **0.9%** ↑ | 91.2% |
| | $\dot{r}$ | **0.9%** ↑ | 86.4% | | $\dot{r}$ | **1.2%** ↑ | 91.5% |

## B.4 TRAINING DETAILS

**CartPole and Pendulum**    The policies use the default network from keras-rl framework. which is a five-layer fully connected network[6]. There are three hidden layers, each of which has 16 units and followed by a rectified nonlinearity. The last output layer is activated by the linear function. For

---

[6]https://github.com/keras-rl/keras-rl/examples

CartPole, We trained the models using Adam optimizer with the learning rate of $1e^{-3}$ for 10,000 steps. The exploration strategy is Boltzmann policy. For DQN and Dueling-DQN, the update rate of target model and the memory size are $1e^{-2}$ and $50,000$. For Pendulum, We trained DDPG and NAF using Adam optimizer with the learning rate of $5e^{-4}$ for $150,000$ steps. the update rate of target model and the memory size are $1e^{-3}$ and $100,000$.

**Atari Games**   We adopt the pre-processing steps as well as the network architecture from Mnih et al. (2015). Specifically, the input to the network is $84 \times 84 \times 4$, which is a concatenation of the last 4 frames and converted into $84 \times 84$ gray-scale. The network comprises three convolutional layers and two fully connected layers[7]. The kernel size of three convolutional layer are $8 \times 8$ with stride 4 (32 filters), $4 \times 4$ with stride 2 (64 filters) and $3 \times 3$ with stride 1 (64 filters), respectively. Each hidden layer is followed by a rectified nonlinearity. Except for Pong where we train the policies for $3e^7$ steps, all the games are trained for $5e^7$ steps with the learning rate of $3e^{-4}$. Note that the rewards in the Atari games are discrete and clipped into $\{-1, 0, 1\}$. Except for Pong game, in which $r = -1$ means missing the ball hit by the adversary, the agents in other games attempt to get higher scores in the episode with binary rewards 0 and 1.

## C   ESTIMATION OF CONFUSION MATRICES

### C.1   REWARD ROBUST RL ALGORITHMS

As stated in Section 3.3, the confusion matrix can be estimated dynamically based on the aggregated answers, similar to previous literature in supervised learning (Khetan et al., 2017). To get a concrete view, we take $Q$-Learning for an example, and the algorithm is called *Reward Robust Q-Learning* (Algorithm 3). Note that is can be extended to other RL algorithms by plugging confusion matrix estimation steps and the computed surrogate rewards, as shown in the experiments (Figure 6).

---

**Algorithm 3** Reward Robust $Q$-Learning

---

**Input:**
  $\tilde{\mathcal{M}} = (\mathcal{S}, \mathcal{A}, \tilde{\mathcal{R}}, \mathcal{P}, \gamma)$: MDP with corrupted reward channel
  $T$: transition function $T : \mathcal{S} \times \mathcal{A} \rightarrow \mathcal{S}$
  $N \in \mathbb{N}$: upper bound of collected noisy rewards
  $\alpha \in (0, 1)$: learning rate in the update rule
  $\eta \in (0, 1)$: weight of unbiased surrogate reward
  $\tilde{R}(s, a)$: set of observed rewards when the state-action pair is $(s, a)$.
**Output:**  $Q(s)$: value function; $\pi(s, t)$: policy function
  Initialize $Q$: $\mathcal{S} \times \mathcal{A} \rightarrow \mathbb{R}$ arbitrarily
  Set confusion matrix $\tilde{\mathbf{C}}$ as zero
  **while** $Q$ is not converged **do**
    Start in state $s \in \mathcal{S}$
    **while** $s$ is not terminal **do**
      Calculate $\pi$ according to $Q$ and exploration strategy
      $a \leftarrow \pi(s)$
      $s' \leftarrow T(s, a)$
      Observe noisy reward $\tilde{r}(s, a)$ and add it to $\tilde{R}(s, a)$
      **if** $\sum_{(s,a)} |\tilde{R}(s, a)| \geq N$ **then**
        Get predicted true reward $\bar{r}(s, a)$ using majority voting in every $\tilde{R}(s, a)$
        Estimate confusion matrix $\tilde{\mathbf{C}}$ based on $\tilde{r}(s, a)$ and $\bar{r}(s, a)$ (Eqn. (4))
        Empty all the sets of observed rewards $\tilde{R}(s, a)$
      Obtain surrogate reward $\hat{r}(s, a)$ using $\mathbf{R}_{proxy} = (1 - \eta) \cdot \mathbf{R} + \eta \cdot \mathbf{C}^{-1} \mathbf{R}$
      $Q(s', a) \leftarrow (1 - \alpha) \cdot Q(s, a) + \alpha \cdot (\hat{r}(s, a) + \gamma \cdot \max_{a'} Q(s', a'))$
      $s \leftarrow s'$
    **return** $Q(s)$ and $\pi(s)$

---

[7]https://github.com/openai/baselines/tree/master/baselines/common

## C.2 EXPECTATION-MAXIMIZATION IN ESTIMATION

In Algorithm 3, the predicted true reward $\bar{r}(s, a)$ is derived from majority voting in collected noisy sets $\tilde{R}(s, a)$ for every state-action pair $(s, a) \in \mathcal{S} \times \mathcal{A}$, which is a simple but efficient way of leveraging the expectation of aggregated rewards without assumptions on prior distribution of noise. In the following, we adopt standard Expectation-Maximization (EM) idea in the our estimation framework (arguably a simple version of it), inspired by previous works (Zhang et al., 2014).

Assuming the observed noisy rewards are independent conditional on the true reward, we can compute the posterior probability of true reward from the Bayes' theorem:

$$
\begin{aligned}
\mathbb{P}(r = R_i | \tilde{r}(1) = R_1, \cdots, \tilde{r}(n) = R_n) &= \frac{\mathbb{P}(\tilde{r}(1) = R_1, \cdots, \tilde{r}(n) = R_n | r = R_i) \cdot \mathbb{P}(r = R_i)}{\sum_j \mathbb{P}(\tilde{r}(1) = R_1, \cdots, \tilde{r}(n) = R_n | r = R_j) \cdot \mathbb{P}(r = R_j)} \\
&= \frac{\mathbb{P}(r = R_i) \cdot \prod_{k=1}^n \mathbb{P}(\tilde{r}(k) = R_k | r = R_i)}{\sum_j [\mathbb{P}(r = R_j) \cdot \prod_{k=1}^n \mathbb{P}(\tilde{r}(k) = R_k | r = R_j)]}
\end{aligned} \tag{8}
$$

where $\mathbb{P}(r = R_j)$ is the prior of true rewards, and $\mathbb{P}(\tilde{r} = R_k | r = R_j)$ is estimated by current estimated confusion matrix $\tilde{\mathbf{C}}$: $\mathbb{P}(\tilde{r} = R_k | r = R_j) = \tilde{c}_{j,i}$. Note that the inference should be conducted for each state-action pair $(s, a) \in \mathcal{S} \times \mathcal{A}$ in every iteration, *i.e.*, $\mathbb{P}(r(s, a) = R_i | \tilde{r}(s, a, 1) = R_1, \cdots, \tilde{r}(s, a, n) = R_n)$, abbreviated as $\mathbb{P}(\bar{r}(s, a) = R_i)$, which requires relatively greater computation costs compared to the majority voting policy. It also points out an interesting direction to check online EM algorithms for our *perturbed-RL* problem.

After the inference steps in Eqn. (8), the confusion matrix $\tilde{\mathbf{C}}$ is then updated based on the posterior probabilities:

$$
\tilde{c}_{i,j} = \frac{\sum_{(s,a)} \mathbb{P}(\bar{r}(s, a) = R_i) \cdot \# [\tilde{r}(s, a) = R_j | \bar{r}(s, a) = R_i]}{\sum_{(s,a)} \mathbb{P}(\bar{r}(s, a) = R_i) \cdot \# [\bar{r}(s, a) = R_i]}, \tag{9}
$$

where $\mathbb{P}(\bar{r}(s, a) = R_i)$ denotes the inference probabilities of true rewards based on collected noisy rewards sets $\tilde{R}(s, a)$. To utilize EM algorithms in the robust reward algorithms (e.g., Algorithm 3), we need to replace Eqn. (4) by Eqn. (9) for the estimation of reward confusion matrix.

## C.3 STATE-DEPENDENT PERTURBED REWARD

In previous sections, to let our presentation stay focused, we consider the state-independent perturbed reward environments, which share the same confusion matrix for all states. In other words, the noise for different states is generated within the same distribution. More generally, the generation of $\tilde{r}$ follows a certain function $C : \mathcal{S} \times \mathcal{R} \to \tilde{R}$, where different states may correspond to varied noise distributions (also varied confusion matrices). However, our algorithm is still applicable except for maintaining different confusion matrices $\mathbf{C}_s$ for different states. It is worthy to notice that Theorem 1 holds because the surrogate rewards produce an unbiased estimation of true rewards for each state, *i.e.*, $\mathbb{E}_{\tilde{r}|r,s_t}[\hat{r}(s_t, a_t, s_{t+1})] = r(s_t, a_t, s_{t+1})$. Furthermore, Theorem 2 and 3 can be revised as:

**Theorem 4.** *(Upper bound) Let $r \in [0, R_{\max}]$ be bounded reward, $\mathbf{C}_s$ be invertible reward confusion matrices with $\det(\mathbf{C}_s)$ denoting its determinant. For an appropriate choice of $m$, the Phased Q-Learning algorithm calls the generative model $G(\hat{\mathcal{M}})$*

$$
O \left( \frac{|\mathcal{S}||\mathcal{A}|T}{\epsilon^2 (1 - \gamma)^2 \min_{s \in \mathcal{S}} \{\det(\mathbf{C}_s)\}^2} \log \frac{|\mathcal{S}||\mathcal{A}|T}{\delta} \right)
$$

*times in $T$ epochs, and returns a policy such that for all state $s \in \mathcal{S}$, $|V_\pi(s) - V^*(s)| \le \epsilon, \epsilon > 0$, w.p. $\ge 1 - \delta$, $0 < \delta < 1$.*

**Theorem 5.** *Let $r \in [0, R_{\max}]$ be bounded reward and all confusion matrices $\mathbf{C}_s$ are invertible. Then, the variance of surrogate reward $\hat{r}$ is bounded as follows:*

$$
\mathbf{Var}(r) \le \mathbf{Var}(\hat{r}) \le \frac{M^2}{\min_{s \in \mathcal{S}} \{\det(\mathbf{C}_s)\}^2} \cdot R_{\max}^2.
$$

### C.4 VARIANCE REDUCTION IN ESTIMATION

As illustrated in Theorem 3, our surrogate rewards introduce larger variance while conducting unbiased estimation which are likely to decrease the stability of RL algorithms. Apart from the linear combination idea (appropriate trade-off), some variance reduction techniques in statistics (e.g., correlated sampling) can also be applied into our surrogate rewards. Specially, Romoff et al. (2018) proposed to a reward estimator to compensate for stochastic corrupted reward signals. It is worthy to notice that their method is designed for variance reduction under stochastic (zero-mean) noise, which is no longer efficacious in more general *perturbed-reward* setting. However, it is potential to integrate their method with our *robust-reward* RL framework because surrogate rewards guarantee unbiasedness in reward expectation.

To verify this idea, we repeated the experiments of *Cartpole* in Section 4.2 but included variance reduction step for estimated surrogate rewards. Following Romoff et al. (2018), we adopted sample mean as a simple approximator during the training and set sequence length as $100$. As shown in Figure 5, the models with only variance reduction technique (red lines) suffer from huge biases when the noise is large, and cannot converge to the optimal policies like those under noisy rewards. Nevertheless, they benefits from variance reduction for surrogate rewards (purple lines), which achieve faster convergence or better performance in many cases (e.g., Figure 5a ($\omega = 0.7$), 5b ($\omega = 0.3$)). It is also not surprising that the integrated algorithm (purple lines) outperforms better as the noise rate increases (indicating larger variance from Theorem 3, e.g., $\omega = 0.9$). Similarly, Table 5 provides quantitative results which show that our surrogate benefits from variance reduction techniques ("ours + VRT"), especially when the noise rate is large.

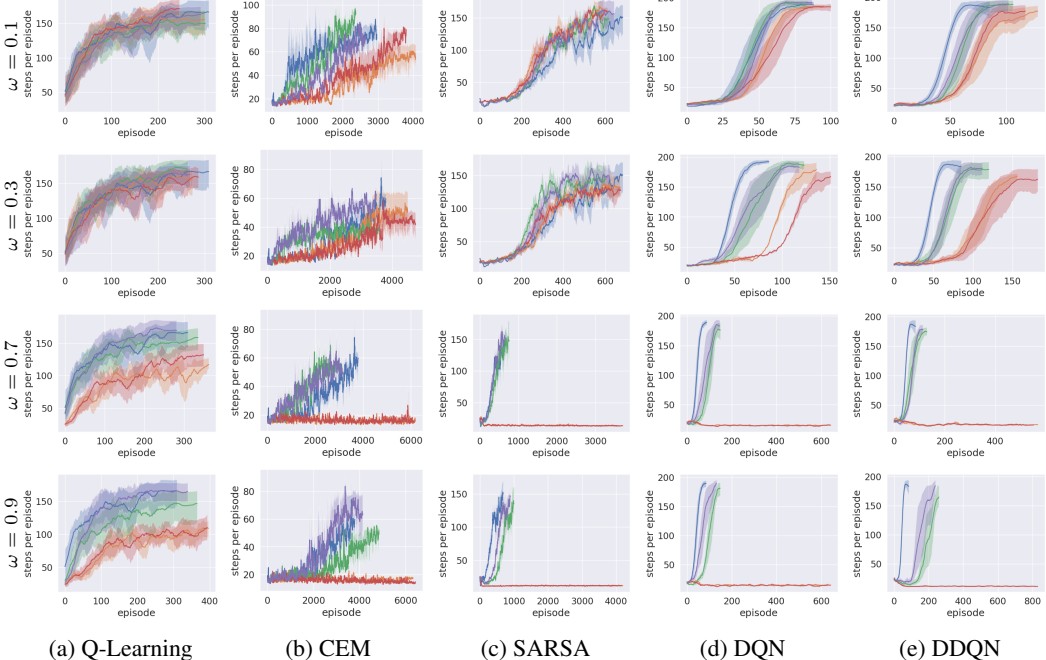

Figure 5: Learning curves from five *reward robust* RL algorithms (see Algorithm 3) on CartPole game with true rewards ($r$) ■, noisy rewards ($\tilde{r}$) ($\eta = 1$) ■, sample-mean noisy rewards ($\eta = 1$) ■, estimated surrogate rewards ($\dot{r}$) ■ and sample-mean estimated surrogate rewards ■. Note that confusion matrices **C** are unknown to the agents here. From top to the bottom, the noise rates are 0.1, 0.3, 0.7 and 0.9. Here we repeated each experiment 10 times with different random seeds and plotted 10% to 90% percentile area with its mean highlighted.

### C.5 EXPERIMENTAL RESULTS

To validate the effectiveness of *robust reward* algorithms (like Algorithm 3), where the noise rates are unknown to the agents, we conduct extensive experiments in *CartPole*. It is worthwhile to notice that the noisy rates are unknown in the explorations of RL agents. Besides, we discretize the

Table 5: Average scores of various RL algorithms on CartPole with sample-mean reward using variance reduction technique (VRT), surrogate rewards (ours) and the combination of them (ours + VRT). Note that the reward confusion matrices are unknown to the agents and the experiments are repeated three times with different random seeds.

| Noise Rate | Reward | Q-Learn | CEM | SARSA | DQN | DDQN |
|---|---|---|---|---|---|---|
| | VRT | 173.5 | **99.7** | 167.3 | 181.9 | **187.4** |
| $\omega = 0.1$ | ours ($\dot{r}$) | 181.9 | 99.3 | 171.5 | **200.0** | 185.6 |
| | ours + VRT | **184.5** | 98.2 | **174.2** | 199.3 | 186.5 |
| | VRT | 140.4 | 43.9 | 149.8 | 182.7 | 177.6 |
| $\omega = 0.3$ | ours ($\dot{r}$) | 161.1 | 81.8 | 159.6 | 186.7 | **200.0** |
| | ours + VRT | **161.6** | **82.2** | **159.8** | **188.4** | 198.2 |
| | VRT | 71.1 | 16.1 | 13.2 | 15.6 | 14.7 |
| $\omega = 0.7$ | ours ($\dot{r}$) | 172.1 | **83.0** | 174.4 | 189.3 | 191.3 |
| | ours + VRT | **182.3** | 79.5 | **178.9** | **195.9** | **194.2** |

observation (velocity, angle, etc.) to construct a set of states and implement like Algorithm 3. The $\eta$ is set 1.0 in the experiments.

Figure 6 provides learning curves from five algorithms with different kinds of rewards. The proposed estimation algorithms successfully obtain the approximate confusion matrices, and are robust in the unknown noise environments. From Figure 7, we can observe that the estimation of confusion matrices converges very fast. The results are inspiring because we don't assume any additional knowledge about noise or true reward distribution in the implementation.

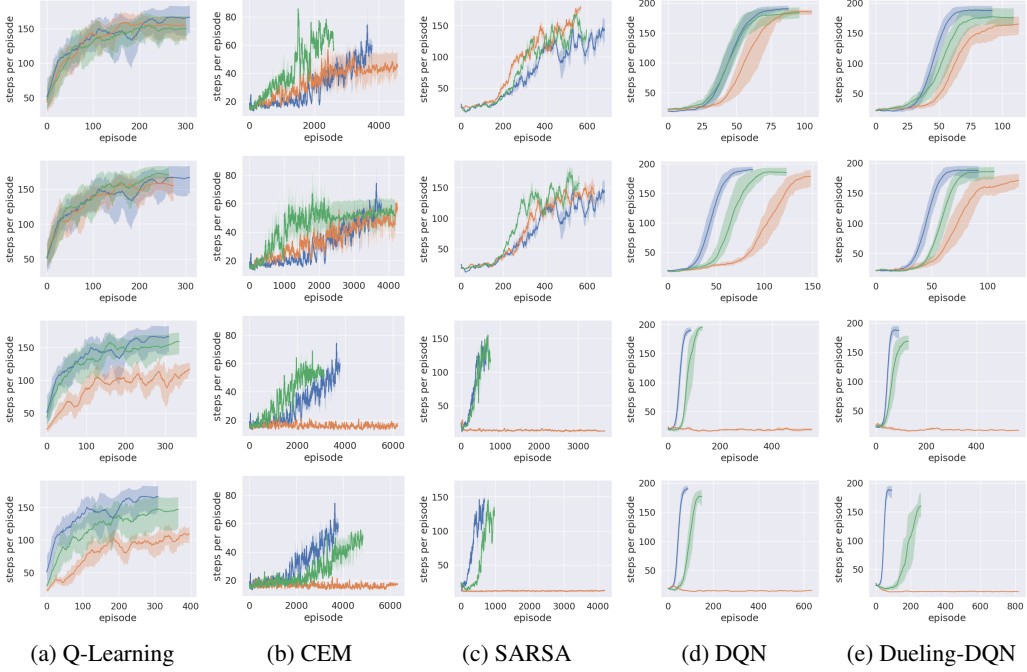

| (a) Q-Learning | (b) CEM | (c) SARSA | (d) DQN | (e) Dueling-DQN |

Figure 6: Complete learning curves from five *reward robust* RL algorithms (see Algorithm 3) on CartPole game with true rewards ($r$) ▇, noisy rewards ($\tilde{r}$) ($\eta = 1$) ▇ and estimated surrogate rewards ($\dot{r}$) ▇. Note that confusion matrices **C** are unknown to the agents here. From top to the bottom, the noise rates are 0.1, 0.3, 0.7 and 0.9. Here we repeated each experiment 10 times with different random seeds and plotted 10% to 90% percentile area with its mean highlighted.

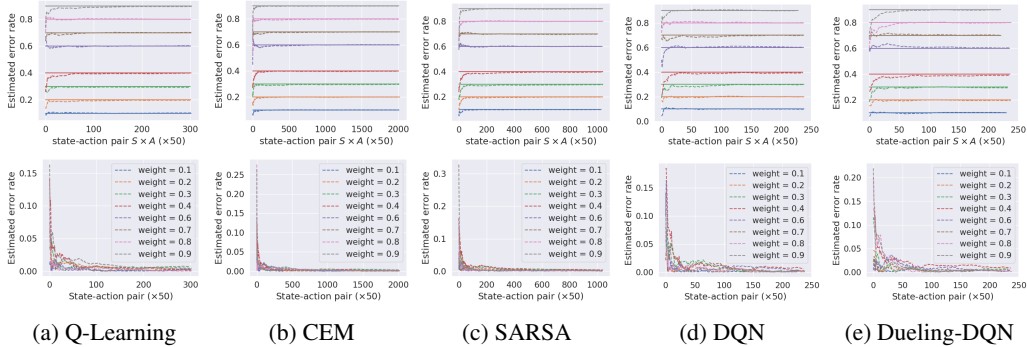

(a) Q-Learning    (b) CEM    (c) SARSA    (d) DQN    (e) Dueling-DQN

Figure 7: Estimation analysis from five *reward robust* RL algorithms (see Algorithm 3) on CartPole game. The upper figures are the convergence curves of estimated error rates (from 0.1 to 0.9), where the solid and dashed lines are ground truth and estimation, respectively; The lower figures are the absolute difference between the estimation and ground truth of confusion matrix $\mathbf{C}$ (normalized matrix norm).

# D SUPPLEMENTARY EXPERIMENTAL RESULTS

## D.1 SUPPLEMENTARY QUANTITATIVE RESULTS

Table 6: Complete average scores of various RL algorithms on CartPole and Pendulum with noisy rewards ($\tilde{r}$) and surrogate rewards under known ($\hat{r}$) or estimated ($\dot{r}$) confusion matrices. Note that the results for last two algorithms DDPG (rand-one) & NAF (rand-all) are on Pendulum, but the others are on CartPole. The experiments are repeated three times with different random seeds.

| Noise Rate | Reward | Q-Learn | CEM | SARSA | DQN | DDQN | DDPG | NAF |
|---|---|---|---|---|---|---|---|---|
| $\omega = 0.1$ | $\tilde{r}$ | 170.0 | 98.1 | 165.2 | 187.2 | **187.8** | -1.03 | -4.48 |
|  | $\hat{r}$ | 165.8 | **108.9** | **173.6** | **200.0** | 181.4 | **-0.87** | **-0.89** |
|  | $\dot{r}$ | **181.9** | 99.3 | 171.5 | **200.0** | 185.6 | -0.90 | -1.13 |
| $\omega = 0.3$ | $\tilde{r}$ | 134.9 | 28.8 | 144.4 | 173.4 | 168.6 | -1.23 | -4.52 |
|  | $\hat{r}$ | 149.3 | **85.9** | 152.4 | 175.3 | 198.7 | **-1.03** | **-1.15** |
|  | $\dot{r}$ | **161.1** | 81.8 | **159.6** | **186.7** | **200.0** | -1.05 | -1.36 |
| $\omega = 0.7$ | $\tilde{r}$ | 56.6 | 19.2 | 12.6 | 17.2 | 11.8 | -8.76 | -7.35 |
|  | $\hat{r}$ | **177.6** | **87.1** | 151.4 | 185.8 | **195.2** | **-1.09** | **-2.26** |
|  | $\dot{r}$ | 172.1 | 83.0 | **174.4** | **189.3** | 191.3 | – | – |

Table 7: Complete average scores of PPO on five selected Atari games with noisy rewards ($\tilde{r}$) and surrogate rewards ($\hat{r}$). The experiments are repeated three times with different random seeds.

| Noise Rate | Reward | Lift (↑) | Mean | Alien | Carnival | Phoenix | MsPacman | Seaquest |
|---|---|---|---|---|---|---|---|---|
| $e_- = e_+ = 0.1$ | $\tilde{r}$ | – | 2044.2 | **1814.8** | 1239.2 | 4608.9 | 1709.1 | 849.2 |
|  | $\hat{r}$ | **67.5%↑** | **3423.1** | 1741.0 | **3630.3** | **7586.3** | **2547.3** | **1610.6** |
| $e_- = 0.1, e_+ = 0.3$ | $\tilde{r}$ | – | 770.5 | 893.3 | 841.8 | 250.7 | 1151.1 | 715.7 |
|  | $\hat{r}$ | **20.3%↑** | **926.6** | **973.7** | **955.2** | **643.9** | **1307.1** | **753.1** |
| $e_- = e_+ = 0.3$ | $\tilde{r}$ | – | 1180.1 | 543.1 | 919.8 | 2600.3 | 1109.6 | **727.8** |
|  | $\hat{r}$ | **46.7%↑** | **1730.8** | **1637.7** | **966.1** | **4171.5** | **1470.2** | 408.6 |
| $e_- = e_+ = 0.7$ | $\tilde{r}$ | – | 296.8 | 485.4 | 380.3 | 126.5 | 491.5 | 0.0 |
|  | $\hat{r}$ | **557.3%↑** | **1951.0** | **1799.2** | **1045.2** | **4970.4** | **1447.8** | **492.5** |
| $e_- = 0.9, e_+ = 0.7$ | $\tilde{r}$ | – | 382.6 | 410.2 | 67.0 | 174.1 | 620.1 | 641.7 |
|  | $\hat{r}$ | **106.9%↑** | **791.5** | **693.5** | **918.1** | **298.9** | **1312.0** | **735.1** |
| $e_- = e_+ = 0.9$ | $\tilde{r}$ | – | 588.8 | 540.6 | 6.3 | 1410.8 | 535.4 | 588.8 |
|  | $\hat{r}$ | **482.3%↑** | **3428.8** | **1901.3** | **4261.7** | **6758.6** | **2515.1** | **1707.1** |

## D.2 VISUALIZATIONS ON CONTROL GAMES

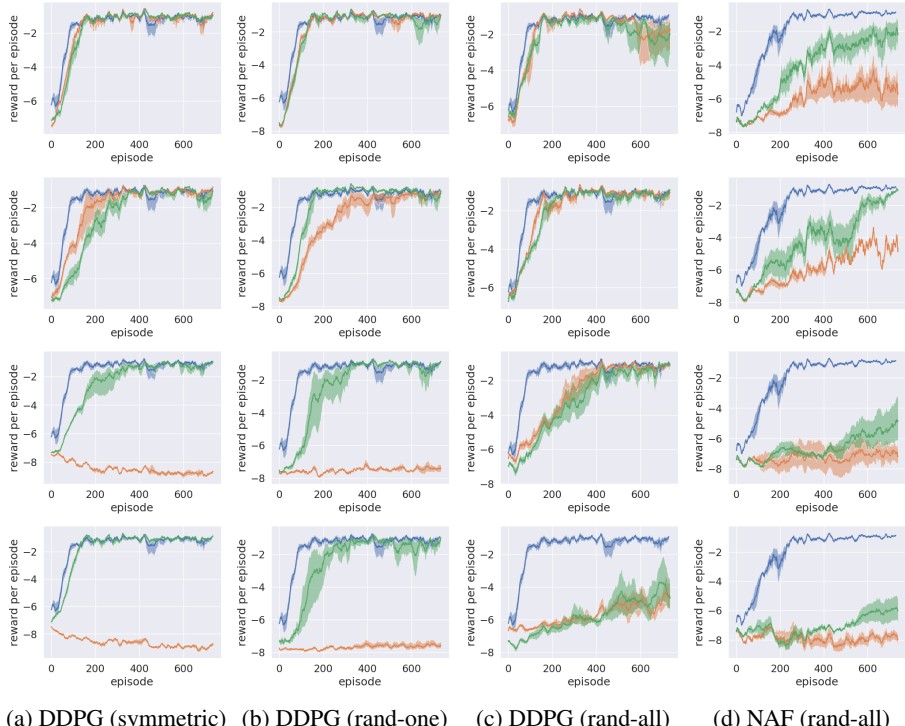

(a) DDPG (symmetric)  (b) DDPG (rand-one)  (c) DDPG (rand-all)  (d) NAF (rand-all)

Figure 8: Complete learning curves from DDPG and NAF on Pendulum game with true rewards $(r)$ ■, noisy rewards $(\tilde{r})$ ■ and surrogate rewards $(\hat{r})$ $(\eta = 1)$ ■. Both symmetric and asymmetric noise are conduced in the experiments. From top to the bottom, the noise rates are 0.1, 0.3, 0.7 and 0.9, respectively. Here we repeated each experiment 6 times with different random seeds and plotted 10% to 90% percentile area with its mean highlighted.

## D.3 VISUALIZATIONS ON ATARI GAMES[8]

### D.3.1 PONG

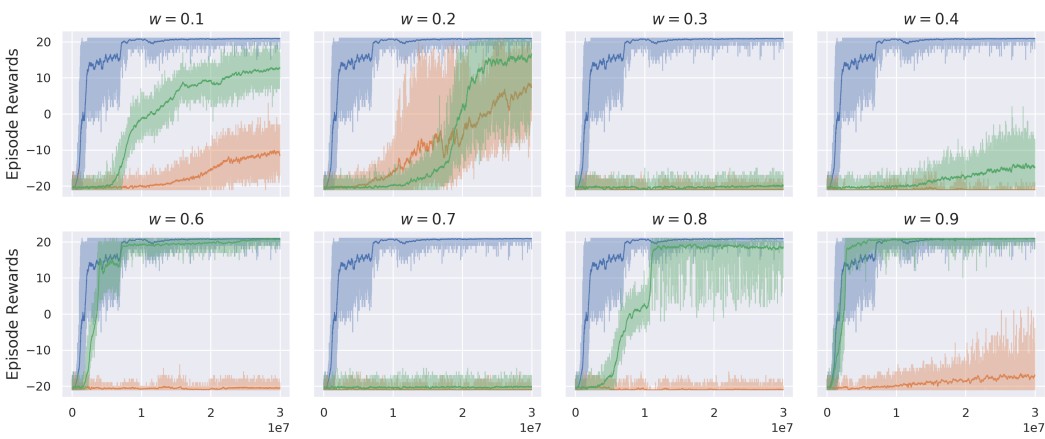

Asymmetric Noise (*rand-one*)

---

[8]For the clarity purpose, we remove the learning curves (blue ones in previous figures) with true rewards except for Pong-v4 game.

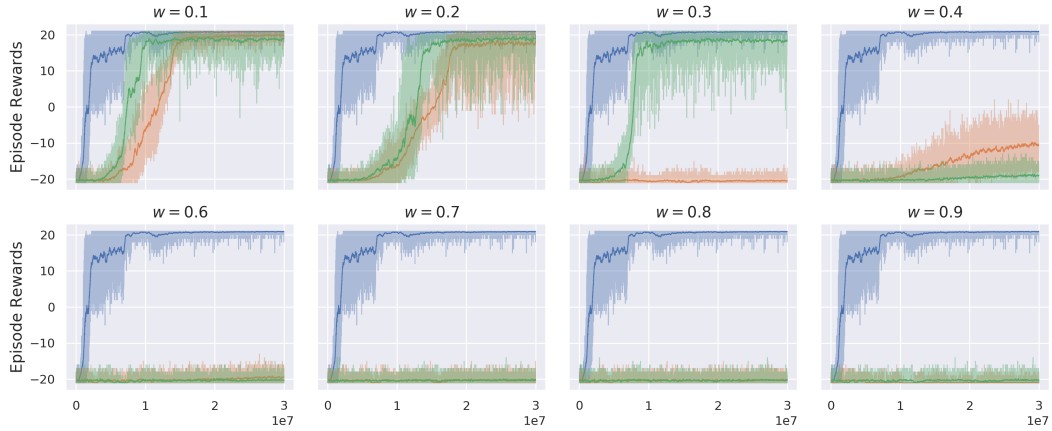

Asymmetric Noise (*rand-all*)

### D.3.2 AIRRAID

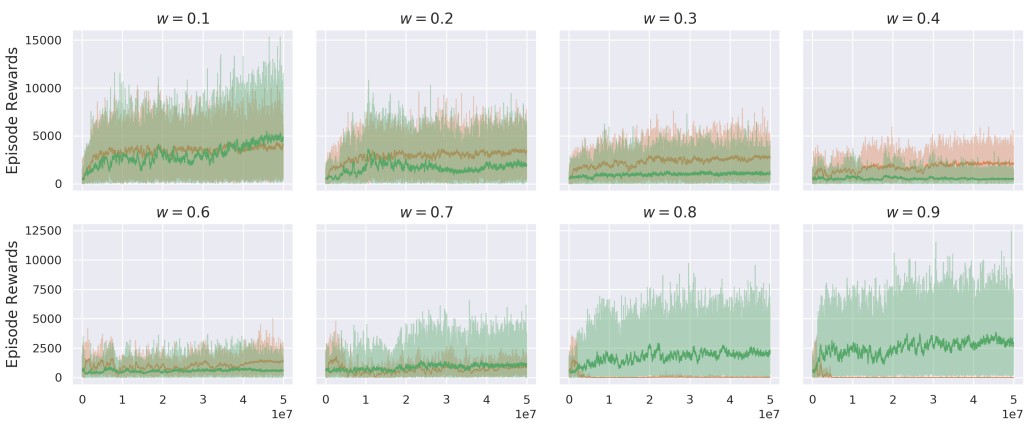

### D.3.3 ALIEN

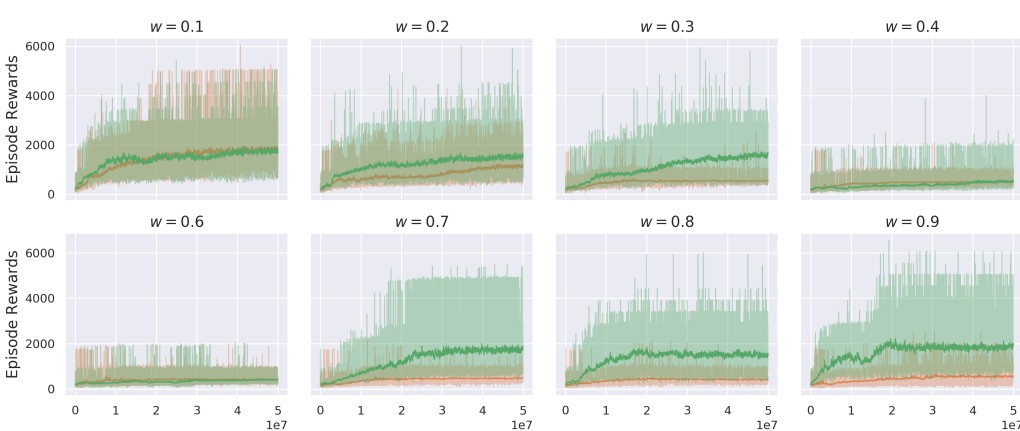

### D.3.4 CARNIVAL

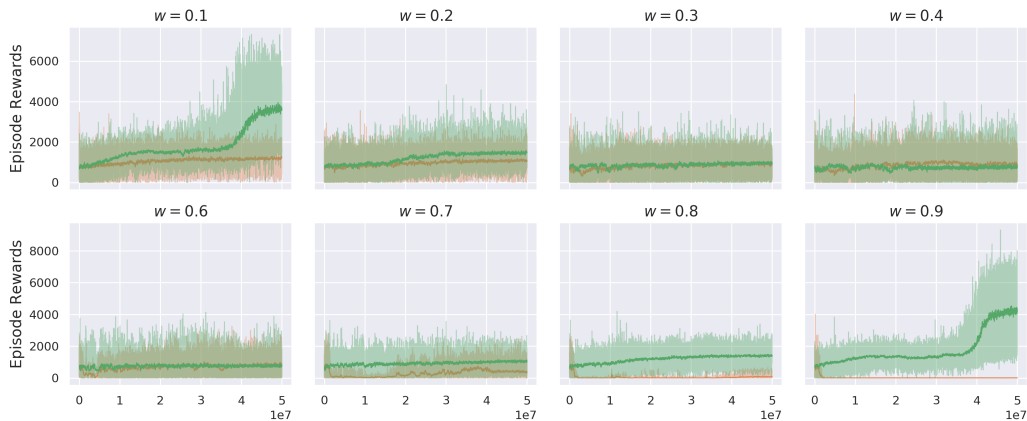

### D.3.5 MSPACMAN

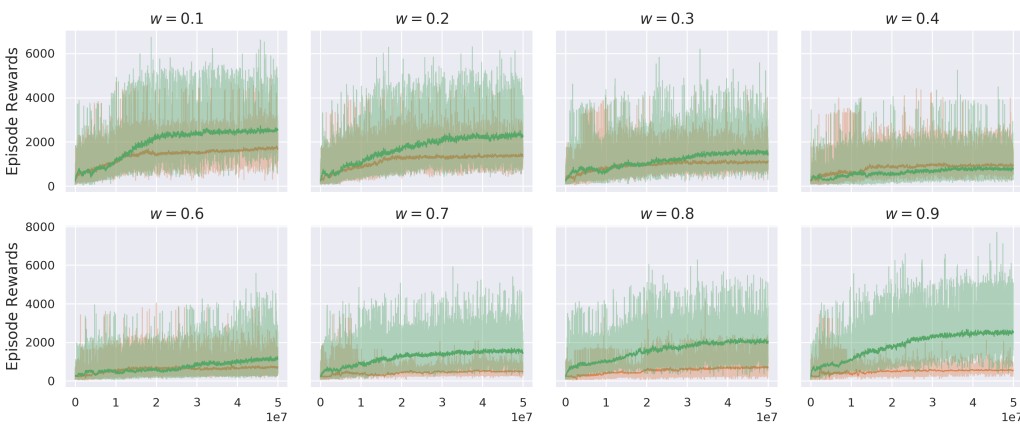

### D.3.6 PHOENIX

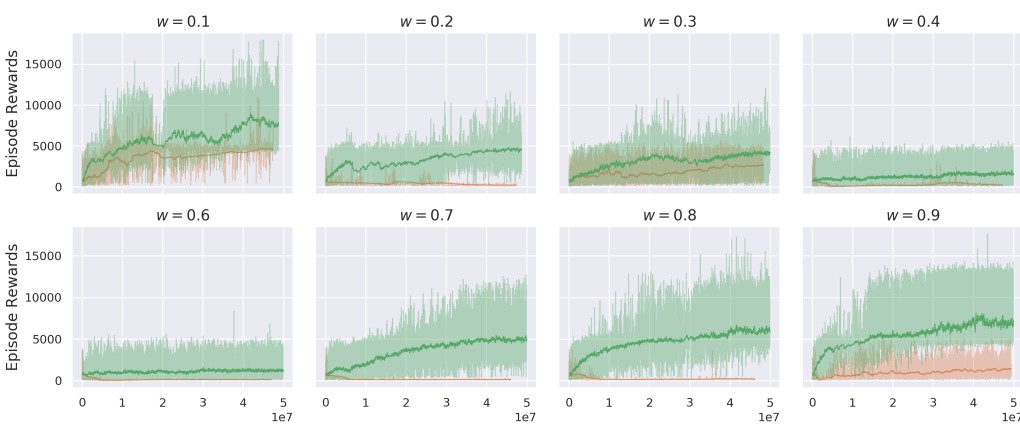

### D.3.7   SEAQUEST

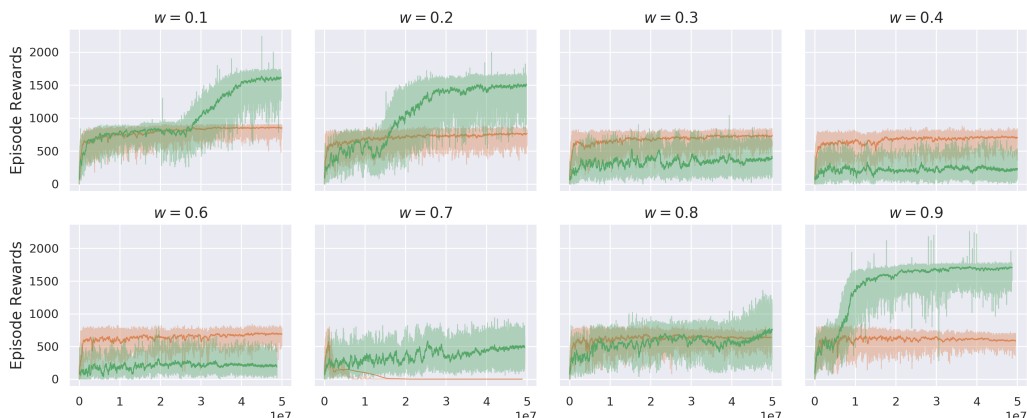

