# OpenReview forum: "Reinforcement Learning with Perturbed Rewards"
_ICLR.cc/2019/Conference_

### Official Review · AnonReviewer3 · 2018-11-02
**interesting but seems to tackle a too narrow problem**

**Rating:** 6
**Confidence:** 3

**Review:**

The paper aims at studying the setting of perturbed rewards in a deep RL setting. Studying the effect of noise in the reward function is interesting. The paper is quite well-written. However the paper studies a rather simple setting, the limitations could be discussed more clearly and there are one or two elements unclear (see below).

The paper assumes first the interesting case where the generation of the perturbed reward is a function of S*R into the perturbed reward space. But then the confusion matrix does *not* take into account the state, which is justified by "to let our presentation stay focused (...)". I believe these elements should at least be clearly discussed. Indeed, in that setting, the theorems given seem to be variations of existing results and it is difficult to understand what is the message behind the theorems.

In addition, it is assumed that the confusion matrix C is known or estimated from data but it's not clear to me how this can be done in practice.  In equation 4, how do you have access to the predicted true rewards?

Additional comments:
- The discount factor can be 0 but can not, in general, be equal to 1. So the equation in paragraph 2.1 "0 < γ ≤ 1" is wrong.
- The paper mention that "an underwhelming amount of reinforcement learning studies have focused on the settings with perturbed and noisy rewards" but there are some works on the subject (e.g., https://arxiv.org/abs/1805.03359) and a discussion about the differences with the related work would be interesting.

---

> ### Author Response · Authors · 2018-11-20
> **Response to AnonReviewer3 -- Part1**
>
> Thanks for your valuable suggestions and comments.
>
> Q1: confusion matrix does not take into account the state
>
> A1: For clarity, our updated draft will stay focusing on the state-independent model, but we have added discussions on the more general cases for the state-dependent noises as suggested (Appendix C.3). When the flipping error rates are different across different states, we will maintain different confusion matrices for each state. More specifically, when a noisy copy of rewards at state $s$ is observed, we look up the corresponding confusion matrix at this state and apply surrogate rewards based on it. In this case, Theorems 1, 2, 3 still hold, with replacing $|\mathbf{C}|$ by $min_s|\mathbf{C}|$. This is because keeping separate confusion matrices for each state could keep the unbiasedness of aggregated rewards (converge to true reward expectation) for each state.
>
> Q2: tackle a narrow problem (the model is simple)
>
> A2: We kindly remind that our model and algorithms deal with a wide range of problems in RL compared to previous works which lie on prior knowledge (Roy et al., 2017) or constraints on noisy distribution such as Gaussian distribution (Romoff et al., 2018). Besides, it is also the first method to estimate confusion matrices in RL settings. The theorems guarantee the convergence and sample efficiency of the proposed method.
>
> We also want to emphasize that our method could be directly applied to various DRL algorithms/environments as our experimental results suggest. In some cases, the surrogate rewards even lead to faster convergence or better scores than the ideal noise-free settings - we conjecture this is because our surrogate reward introduces implicit exploration via noise (followed by the debiasing ste). We believe that our “perturbed reward” setting and the algorithms are of both theoretical and practice values to the RL community.
>
> Q3: theorem seems to be variations of existing results; difficult to understand what is the message behind the theorems.
>
> A3: Indeed, building on existing results, our theorems are not surprising. This is also mentioned by ourselves in the paper. The theorems in the paper are established in sequence to provide theory guarantee (convergence, sample complexity, and variance) of the proposed unbiased estimator.
> More specially, theorem 1 states our surrogate rewards retain the convergence property of Q-Learning; theorem 2 shows the sample efficiency of Phased Q-Learning under surrogate rewards – bounded by an extra constant factor det|\mathbf{C}| than true rewards; theorem 3 discusses the limitation that surrogate rewards introduce larger variance compared to the cases that observe true rewards.
>
> Q4: Not clear how confusion matrix can be estimated in practice; how to access predicted true rewards
>
> A4: The procedure of estimating confusion matrix C is given in Algorithm 1 and Equation 4. Briefly speaking, at each time step, we first obtain the predicted rewards (using majority voting, but we have added more discussions on using other inference methods, e.g. EM (Appendix C.2), in our updated version) for each state-action pair. In practice, we would discretize the state and action if it is continuous (e.g., Pendulum). Then we estimate the confusion matrix C using Equation 4 and use surrogate rewards for training. Extensive experiments conducted on various algorithms and environments show that the proposed estimation algorithm can obtain a significant improvement in overall scores (\dot{r} in Table1 and Table2); especially this helps rescue agents from misleading rewards when the noise is high. It could also be observed from Figure 5 that the estimations of the confusion matrices converge to the true ones reasonably fast. Note that our algorithm is generic and efficient in practice, which can be flexibly plugged in any existing RL algorithm.

---

> > ### Author Response · Authors · 2018-11-20
> > **Response to AnonReviewer3 -- Part2**
> >
> > Response to additional comments:
> >
> > Thanks for your suggestion. We added the comparisons with previous work (Romoff et al., 2018) and analysis (Appendix C.4). Briefly, Romoff et al. focused more the variance reduction issue, which theoretically doesn’t resolve the challenge when bias presents in the observed rewards. While our study sets out to deal with a reward model with bias. Indeed the idea from (Romoff et al., 2018) can be used as a second variance reduction step following our surrogate reward operation (which unavoidably introduced higher variance due to the bias removal step). We have conducted experiments to show its further benefits.
> >
> > For the discount factor gamma, we considered both the discounted ($0 \leq \gamma < 1$) and undiscounted MDP ($\gamma = 1$) setting (Schwartz et al., 1993; Sobel et al., 1994; Kakade, 2003). So we took your and Reviewer 4's suggestion, adjusting the range to be [0, 1].
> >
> > [1] A. Roy, H. Xu, and S. Pokutta. Reinforcement learning under Model Mismatch. 31st Conference on Neural Information Processing Systems, 2017.
> > [2] J. Romoff, A. Piché, P. Henderson, V. Francois-Lavet, and J. Pineau. Reward estimation for variance reduction in deep reinforcement learning. ICLR Workshop, 2018.
> > [3] A. Schwartz. A reinforcement learning method for maximizing undiscounted rewards. In ICML, pp. 298–305. Morgan Kaufmann, 1993.
> > [4] M. J. Sobel.  Mean-variance tradeoffs in an undiscounted MDP. Operations Research, 42(1):175–183, 1994.
> > [5] S. M. Kakade.On the Sample Complexity of Reinforcement Learning. PhD thesis, University of London, 2003.

---

> > > ### Comment · AnonReviewer3 · 2018-11-23
> > > **Thanks for the detailed answer**
> > >
> > > Thank you for your detailed response and the update of your paper.
> > >
> > > Since you have added a discussion on a state dependent confusion matrix as well as a discussion and experiments for the related work, I can rise my score to 6 from 5. I believe the way the confusion matrix is estimated from data is still among the relatively weak points to really have a widespread use but this paper has nonetheless its merits.

---

### Official Review · AnonReviewer1 · 2018-11-03
**An interesting general surrogate reward which has wide applicability, and can be flexibly included alongside a variety of algorithms.**

**Rating:** 6
**Confidence:** 3

**Review:**

## Summary

The authors present work that shows how to deal with noise in reward signals by creating a surrogate reward signal. The work develops a number of results including: showing how the surrogate reward is equal in expectation to the true reward signal, how this doesn't affect the fixed point of the Bellman equation, how to deal with finite and continuous rewards and how the convergence time is affected for different levels of noise. They demonstrate the value of this approach with a variety of early and state-of-the-art algorithms on a variety of domains,, and the results are consistent with the claims.

It would be useful to outline how prior work approached this same problem and also to evaluate the proposed method with existin approaches to the same problem. I realise that this is the first method that estimates the confusion matrix rather than assuming it is known a priori but there are obvious ways around this, e.g. the authors first experiment assumes the confusion matrix is known, so this would be a good place to compare with other competing techniques. Also, the authors have a way of estimating this, so they could plug it into the other algorithms too.

I also have some concerns about the clarity and precision of the proofs, although I do not have any reason to doubt the Lemma/Theorem correctness (see below).

The weakest part of the approach is in how the true reward is estimated in order to estiamate the confusion matrix. It uses majority vote (which is only really possible in the case of finite rewards with noise sufficiently low that this will be a robust estimate). Perhaps some other approaches could be explore here too.

Finally, there is discussion about adversarial noise in rewards at the beginning but I am not sure the theory really addresses it nor the evaluations.

Nonetheless, given that I do not know whether the claim of originality is true (in terms of the estimation of the confusion matrix). If it is, then the work is a significant and interesting advance, and is clearly widely applicable in domains with noisy rewards. It would be interesting to see a more tractable approach for continous noise too, but this would probably involve assumptions (smoothness? Gaussianity?), and doesn't impact the value of this work.

## Detailed notes

There is a slight sloppiness in  notation in equation (1). This uses \tilde{r} as a subscript of e, but r is +1 or -1 and the error variables are e_+ and e_- (not e_{+1} and e_{-1}).


The noise levels in Atari (Figure 3) show something quite interesting which could be commented upon. For noise below 0.5 the surrogate reward works roughly  similarly to the noisy reward, but when the noise level goes above this, the surrogate reward clearly exploits the increased information content (similar to a noisy binary channel with over 0.5 noise). This may have  implications for adversarial noise.

There are also some issues with the proofs which I spotted outlined below:

### Lemma 1 proof
The proof of Lemma 1, I think, fails to achieve its objective. The first pair of equations is not a rewrite of equation (1). I believe that the authors intend for this to be a consequence of Equation (1) but do not really demonstrate this clearly. Also, the authors seem to switch between binary rewards -1 and +1 and two levels of reward r- and r+ leading to some confusion. I would suggest the latter throughout as it is more general but involves no more terms.

I suggest the following as an outline for the proof. It would help for them to define what they mean by the different rhats (as they currently do) and explain that these values are therefore:

  rhat- = [(1 - e+) r- - e- r+ ]/(1 - e+ - e-)
  rhat+ = [(1 - e-) r+ - e+ r-]/(1- e+ - e-)

from equation (1). What is left is for them to actually prove the Lemma, namely that the expected value of rhat is:

  E(rhat ) = p1(rhat=rhat-) rhat- + p(rhat=rhat+) rhat+ = E(r)

where the probabilities relate to the surrogate reward taking their respective values. And just stylistically, I would avoid writing "we could obtain" and simply write "we obtain".

Lemma 2 achieves this more clearly with greater generality.


### Theorem 1 proof
At the end of p13, the proof of the expected value loses track of the chosen action a. I would suggest the authors replace: $$\mathbb{P}'(s,s',\hat{r})$$ with $$\mathbb{P}'(s,a, s',\hat{r})$$ then define it. Likewise $$\mathbb{P}(s,s')$$ should be $$\mathbb{P}(s,a,s')$$ (and also defined).

I am also a little uncomfortable with the switch from: $$max_{b \in \mathcal{A}} | Q(s',b) - Q*(s',b)|$$ in the second to last line of p13, which refers to the maximum Q value associated with some state s', to  $$||Q-Q*||_{\infty}$$ in the next line which is the maximum over all states and actions. The equality should probably be an inequality there too.

Throughout this the notation could be much better defined, including how to interpret the curly F and how it acts in the conditional part of an expectation and variance.

Finally, there is a bit too free a use of the word "easily" here. If it were easy, then the authors could do it more clearly I think. Otherwise, please refer to the appropriate result in the literature.

---

> ### Author Response · Authors · 2018-11-20
> **Response to AnonReviewer1**
>
> Thanks for your valuable suggestions and comments.
>
> Q1: Compare existing approaches to the same problem; plug it into other algorithms for enhancement.
>
> A1: As also mentioned by the reviewer, our surrogate reward method is the first one that estimates the confusion matrix rather than assuming it is a known prior. Meanwhile, the “perturbed reward” setting, where the noise is generated by confusion matrices, is very different from previous settings for RL with noisy environment (e.g., prior of transition matrices (Roy et al., 2017); Gaussian noise (Romoff et al., 2018)); therefore, it is difficult to compare with the above approaches directly (in some sense these methods target different set of questions, and are rather incomparable). However, we found that it is feasible to compare with Romoff et al.’s work and test their method under the noises generated according to a confusion matrix. We did experiments on Cartpole using various RL algorithms and proved our surrogate estimator based RL algorithms outperform their method consistently (In Appendix C.4).
>
> From Theorem 3, we know that surrogate rewards, though correct biases suffer high variance cost - this newly introduced variance can be reduced using the idea introduced by Romoff et al.’s work. We added relevant discussions and new experimental results in Appendix C.4. We’d like to thank the reviewer for your suggestion!
>
> Q2: "weakest part -- majority voting"
>
> A2: Majority voting is only one of the simple but effective methods for inferring the ground truth, and then further the error rates. Because we are in a sequential setting and because of the fact that agents could only observe one copy of the noisy rewards based on their own explorations, other more sophisticated inference algorithms that were proposed in crowdsourcing literature (which often requires the availability of multiple redundant copies simultaneously) cannot be directly applied. This is a very interesting direction that merits more rigorous future explorations.
>
> Nonetheless, we can adapt standard Expectation-Maximization (EM) idea into our estimation algorithm. We provide the derivation (Appendix C.2) in our updated version. However, it is worthwhile to note that the inference probabilities need to be computed at every iteration, which introduces larger computation costs - this points out an interesting direction of checking online EM algorithms for our RL problem.
>
> Q3: adversarial noise -- no evaluations in the paper
>
> A3: In this paper, we did not address more general cases with arbitrary adversarial noises; we have clarified it in the updated version. But we’d like to emphasize that the proposed estimator deals with a generic setting without any assumptions on the distribution of true reward distribution (which can be fully characterized via the noisy observations and the estimated confusion matrices). Our solution is also robust to time-varying noises. We have added results on this (Figure 4(b) in Appendix B.3). We hope our solution provides a practical baseline for defending adversarial noise in the future work.
>
> Q4: originality of confusion matrix estimation in RL; more tractable approach for continuous noise in the future work
>
> A4: Thanks for sharing your ideas. Our work demonstrates a happy marriage between crowdsourcing and RL. For the case with continuous noise, if not utilizing the discretization method, we agree it is probably necessary to involve assumptions of the distributions to extend to this continuous scenario. It is an interesting and a very practical question to explore in the future, but right now we don’t have more concrete ideas other than using fixed discretization.
>
> Response to detailed suggestions:
>
> Thanks for your valuable suggestions. We have revised the proofs and explanations (Lemma 1, 2 and Theorem 1) in the updated paper according to your suggestions.
>
> Reference:
> [1] A. Roy, H. Xu, and S. Pokutta. Reinforcement learning under Model Mismatch. 31st Conference on Neural Information Processing Systems, 2017.
> [2] J. Romoff, A. Piché, P. Henderson, V. Francois-Lavet, and J. Pineau. Reward estimation for variance reduction in deep reinforcement learning. ICLR Workshop, 2018.

---

### Official Review · AnonReviewer4 · 2018-11-08
**An interesting and relatively unexplored variant of RL.**

**Rating:** 6
**Confidence:** 4

**Review:**


This paper investigates reinforcement learning with a perturbed reward signal. In particular, the paper proposes a particular model for adding noise to the reward function via a confusion matrix, which offers a nuanced notion of reward-noise that is not too complicated so-as to make learning impossible. I take this learning setting to be both novel and interesting for opening up areas for future work. The central contributions of the work are to 1) leverage a simple estimator to prove the convergence of Q-Learning under the reward-perturbed setting along with the sample-complexity of a variant of (Phased) Q-Learning which they call "Phrased" Q-Learning, and 2) An algorithmic scheme for learning in the reward-perturbed setting (Algorithm 1), and 3) An expansive set of experiments that explore the impact of various reward models on learning across different environment-algorithm combinations. The sample complexity term extends Phased Q-Learning to incorporate aspects of the reward confusion matrix, and to my knowledge is novel. Further, even though Theorem 1 is unsurprising (as the paper suggests), I take the collection of Theorem 1, 2, and 3 to be collectively novel.

Indeed, the paper focuses on an interesting and relatively unexplored direction for RL. Apart from the work cited by the paper (and perhaps work like Krueger et al. (2016), in which agents must pay some cost to observe true rewards), there is little work on learning settings of this kind. This paper represents a first step in gaining clarity on how to formalize and study this problem. I did, however, find the analysis and the experiments to be relatively disjointed -- the main sample complexity result presented by the paper (Theorem 2) was given for Phased Q-Learning, yet no experiments actually evaluate the performance of Phased Q-Learning. I think the paper could benefit from experiments focused on simple domains that showcase how traditional algorithms do in cases where it is easier to understand (and visualize) the impact of the reward perturbations (simple chain MDPs, grid worlds, etc.) -- and specifically experiments including Phased Q-Learning.

Pros:
	- General, interesting new learning setting to study.
	- Initial convergence and sample complexity results for this new setting.
	- Depth and breadth of experimentation (in terms of diversity of algorithms and environments), includes lots of detail about the experimental setup.

Cons:
	- Clarity of writing: lots of typos and bits of math that could be more clear (see detailed comments below) [Fixed]
	- The plots in Section 4 are all extremely jagged. More trials seem to be required. Moreover, I do think simpler domains might help offer insights into the reward perturbed setting. [Fixed]
	- The reward perturbation model is relatively simple.

Some high level questions/comments:
	- Why was Phrased Q-Learning not experimented with?
	- Why use majority voting as the rule? When this was introduced it sounded like any rule might be used. Have you tried/thought about others?
	- Your citation to Kakade's thesis needs fixing; it should read:
		"Kakade, Sham Machandranath. On the sample complexity of reinforcement learning. Ph.D Thesis. University of London, 2003."

		(right now it is cited as "(Gatsby 2003)" throughout the paper)
	- You might consider picking a new name for Phrased Q-Learning -- right now the name is too similar to Phased Q-Learning from [Kearns and Singh NIPS 1999].
        - As mentioned in the "cons" section, the confusion matrix is still a somewhat simple model of reward noise. I was left wondering: what might be the next most complicated form of adding reward noise? How might the proposed algorithm(s) respond to this slightly more complex model? That is, it's unclear how general the results are, or if they are honed too tightly to the specific proposed reward noise model. I was hoping the authors could respond to this point.


Section 0) Abstract:
	- Not immediately clear what is meant by "vulnerability" or "noisy settings". Might be better to pick a more clear initial sentence (same can be said of the "sources of noise..."")

Section 1) Introduction:
	- "adversaries in real-world" --> "adversaries in the real-world"
	- You might consider citing Loftin et al. (2014) regarding the bulleted point about "Application-Specific Noise".
	- "unbiased reward estimator aided reward robust reinforcement learning framework" --> this was a bit hard to parse. Consider making more concise, like: "unbiased reward estimator for use in reinforcement learning with perturbed rewards".
	- "Our solution framework builds on existing reinforcement learning algorithms, including the recently developed DRL ones" --> cite these up front So, cite: Q-Learning, CEM, SARSA, DQN, Dueling DQN, DDPG, NAF, and PPO, and spell out the acronym for each the first time you introduce them.
	- "layer of explorations" --> "layer of exploration"

Section 2) Problem Formulation
	- "as each shot of our" --> what is 'shot' in this context?
	- "In what follow," --> "In what follows,"
	- "where 0 < \gamma \leq 1" --> Usually, $\gamma \in [0,1)$, or $[0,1]$. Why can't $\gamma = 0$?
	- The transition notation changes between $\mathbb{P}_a(s_{t+1} | s_t)$ and $\mathbb{P}(s_{t+1} | s_t, a_t)$. I'd suggest picking one and sticking with it to improve clarity.
	- "to learn a state-action value function, for example the Q-function" --> Why is the Q-function just an example? Isn't is *the* state-action value function? That is, I'd suggest replacing "to learn a state-action value function, for example the Q-function" with "to learn a state-action value function, also called the Q-function"
	- "Q-function calculates" --> "The Q-function denotes"
	- "the reward feedbacks perfectly" --> "the reward feedback perfectly"
	- I prefer that the exposition of the perturbed reward MDP be done with C in the tuple. So: $\tilde{M} = \langle \mathcal{S}, \mathcal{A}, \mathcal{R}, C, \mathcal{P}, \gamma \rangle$. This seems the most appropriate definition, since the observed rewards will be generated by $C$.
	- The setup of the confusion matrix for reward noise over is very clean. It might be worth pointing out that $C$ need not be Markovian. There are cases where C is not just a function of $\mathcal{S}$ and $\mathcal{R}$, like the adversarial case you describe early on.


Section 3) Learning w/ Perturbed Rewards
	- Theorem 1 builds straightforwardly on Q-Learning convergence guarantee (it might be worth phrasing the result in those terms? That is: the addition of the perturbed reward does not destroy the convergence guarantees of Q-Learning.)
	- "we firstly" --> "we first"
	- "value iteration (using Q function)" --> "value iteration"
	- "Definition 2. Phased Q-Learning" --> "Definition 2. Phrased Q-Learning". I think? Unless you're talking about Phased Q from the Kearns and Singh '99 work.
	- "It uses collected m samples" --> "It uses the collected m samples"
	- Theorem 2: it would be helpful to define $T$ since it appears in the sample complexity term. Also, I would suggest specifying the domain of $\epsilon$, as you do with $\delta$.
	- "convergence to optimal policy" --> "convergence to the optimal policy"
	- "The idea of constructing MDP is similar to" --> this seems out of place. The idea of constructing which MDP? Similar to Kakade (2003) in what sense?
	- "the unbiasedness" --> "the use of unbiased estimators"
	- "number of state-action pair, which satisfies" --> "number of state-action pairs that satisfy"
	- "The above procedure continues with more observations arriving." --> "The above procedure continues indefinitely as more observation arrives." Also, which procedure? Updating $\tilde{c}_{i,j}$? If so, I would specify.
	- "is nothing different from Eqn. (2) but with replacing a known reward confusion" --> "replaces a known reward confusion"


4) Experiments:
	- Diverse experiments! That's great. Lots of algorithms, lots of environment types.
	- I expected to see Phrased Q-Learning in the experiments. Why was it not included?
	- The plots are pretty jagged, so I'm left feeling a bit skeptical about some of the results. The results would be strengthened if the experiments were repeated for more trials.

5) Conclusion:
	- "despite of the fact" --> "despite the fact"
	- "finite sample complexity of Q-Learning with estimated surrogate rewards are given" --> It's not really Q-Learning, though. It's a variant of Q-Learning. I'd suggest being explicit about that.

Appendix:

	- "It is easy to validate the unbiasedness of proposed estimator directly." --> "It is easy to verify that the proposed estimator is unbiased directly."
	- "For the simplicity of notations" --> "For simplicity"
	- "the Phrased Q-Learning could converge to near optimal policy" --> ""the algorithm Phrased Q-Learning can converge to the near optimal policy""
	- "Using union bound" --> "Using a union bound"
	- Same comment regarding $\gamma$: it's typically $0 \leq \gamma < 1$.
	- Bottom of page 16, the second equation from the bottom, far right term: $c.j$ --> $c,j$.
	- "Using CauchySchwarz Inequality" --> "Using the Cauchy-Schwarz Inequality"


References:
	Loftin, Robert, et al. "Learning something from nothing: Leveraging implicit human feedback strategies." Robot and Human Interactive Communication, 2014 RO-MAN: The 23rd IEEE International Symposium on. IEEE, 2014.

	Krueger, D., Leike, J., Evans, O., & Salvatier, J. (2016). Active reinforcement learning: Observing rewards at a cost. In Future of Interactive Learning Machines, NIPS Workshop.

---

> ### Author Response · Authors · 2018-11-20
> **Response to AnonReviewer4**
>
> Thanks for your valuable suggestions and positive comments.
>
> Response to high level questions/comments:
> Q1: supplement experiments on simpler environments such as simple MDP.
>
> A1: Thanks for the suggestion. We added relevant MDP experiments (Appendix B.3) in the revised paper to provide insights and intuitive understandings of the perturbed reward setting.
>
> Q2: Why was Phased Q-Learning not experimented with?
>
> A2: For phased Q-Learning, it is a theoretical algorithm to help establish the proof of finite sample complexity (Kearns and Singh, 1999; Kakade, 2003), which is not a popular solution adopted in real-world applications. Compared to general Q-Learning algorithms, the Phased Q-Learning is not practical due to its stronger assumptions (taking m samples per phase). Consequently, we tested and present the experimental results for the general Q-Learning.
>
> Q3: Why use majority voting as the rule? Have you tried others?
>
> A3: Majority voting is only one of the simple but effective methods for inferring the ground truth. Because we are in a sequential setting and the fact that agents could only observe one-copy of noisy rewards based on their own explorations, other more sophisticated inferences algorithms that were proposed in crowdsourcing cannot be directly applied. This is a very interesting topic that merits more rigorous future explorations. Nonetheless, we can adapt standard Expectation-Maximization (EM) idea into our estimation algorithm. We provide the derivation (Appendix C.2) in our updated version. However, it is worth noting that the inference probabilities need to be computed in every iteration, which introduces larger computation costs - this points out an interesting direction to check online EM algorithms for our RL problem.
>
> Q4: reward perturbed setting is relatively simple & How might the proposed algorithm(s) respond to this slightly more complex model?
>
> A4: Although the “perturbed reward” setting seems relatively simple, it deals with a wider problem in RL compared to previous works which depend on prior knowledge of the RL environment (Roy et al., 2017) or constraints on noisy distribution such as Gaussian distribution (Romoff et al., 2018). Besides, we provide solutions for continuous noises and corresponding estimation algorithm when the confusion matrices are unknown to the agents. We believe that the more complex cases would be state-dependent noise (different confusion matrices for each state), time-variant noise (confusion matrices are time-variant) and adversarial noise. Our algorithm could handle time-variant noise (shown in Appendix B.3) because the estimated confusion matrices are dynamically updated based on the temporal noisy reward sequence.
>
> Q5: citation needs fixing & “Phased Q-Learning”?
>
> A5: Thanks for pointing it out. There are some typos in the citation and the algorithm’s name. In the updated paper, we fixed the citation and the algorithm name (as “Phased Q-Learning” not “Phrased Q-Learning”).
>
> Response to Cons:
> Q1: Typos and bits of math could be clearer.
>
> A1: Thank you for the meticulous proofreading and valuable comments. We have addressed them carefully in the updated paper.
>
> Q2: Some plots are jagged -- more trials seem to be required.
>
> A2: Thanks for the suggestion. It was a problem of visualization -- we plotted all the curves for all repeated experiments so it looks jagged. We changed the plotting style in the revised paper. Besides, the number of trials also matter as you suggest. In our previous sets of experiments, we repeated each experiment for three times under different random seeds. Due to the fact that some algorithms (e.g., Q-Learning, CEM, SARSA) see high variances while playing OpenAI Gym games, the plots sometimes look jagged. To reach higher statistical significance, we now experimented with more trials (10 times for Cartpole and 6 times for Pendulum) and updated the figures (Figure 1, 2) in the paper.
>
> Reference:
> [1] A. Roy, H. Xu, and S. Pokutta. Reinforcement learning under Model Mismatch. 31st Conference on Neural Information Processing Systems, 2017.
> [2] J. Romoff, A. Piché, P. Henderson, V. Francois-Lavet, and J. Pineau. Reward estimation for variance reduction in deep reinforcement learning. ICLR Workshop, 2018.
> [3] M. J. Kearns, Y. Mansour, and A. Y. Ng. A sparse sampling algorithm for near-optimal planning in large markov decision processes. In IJCAI, pp. 1324–1231. Morgan Kauf-mann, 1999.
> [4] S. M. Kakade.On the Sample Complexity of Reinforcement Learning. PhD thesis, University of London, 2003.

---

> > ### Comment · AnonReviewer4 · 2018-11-25
> > **Response to Detailed Response & Revision**
> >
> > Thank your for your response and the revisions. The revised version of the paper includes many improvements. The results presented in Figures 1, 2, & 3 are much more compelling with the additional trials. I also appreciate the inclusion of the simple MDP example in the appendix, and the editorial changes made; the paper reads much more smoothly now.
> >
> > In light of the updates, I have changed my score from a 6 to a 7.

---

### Public Comment · (anonymous) · 2018-11-08
**Related work on robust RL with perturbed state transitions**

Thank you for your work! A related work on robust RL with the perturbations on the state transitions (rather than the rewards as in your setting) is [1].

[1] https://papers.nips.cc/paper/6897-reinforcement-learning-under-model-mismatch.

---

> ### Author Response · Authors · 2018-11-20
> **Response to "Related work on robust RL with perturbed state transitions"**
>
> Thanks for your interest in our work! It is indeed beneficial for the community to have a more comprehensive view of robust RL results. We now have mentioned the relevant papers on robust RL with model uncertainty (on state transitions) in the related work. Thanks for pointing this out.

---

### Author Response · Authors · 2018-11-20
**Revised manuscript has been uploaded**

We would like to thank the reviewers and the other anonymous comment again for their thoughtful reviews and valuable comments. We have made our best efforts to improve the paper according to the comments. The key changes include adding experiments on simpler domains (MDP example) and time-variant cases (Appendix B.3), comparing with another baseline (Romoff et al., 2018), utilizing variance reduction technique to further improve our performance (Appendix C.4), adapting EM idea into our estimation algorithm (Appendix C.2), clarifying the proofs, fixing the typos and providing more analysis of "perturbed reward" setting (state-dependent case, Appendix C.3).

[1] J. Romoff, A. Piché, P. Henderson, V. Francois-Lavet, and J. Pineau. Reward estimation for variance reduction in deep reinforcement learning. ICLR Workshop, 2018.

---

### Meta-Review · Area_Chair1 · 2018-12-14
**Interesting direction but contributions and significance somewhat limited**

**Confidence:** 4
**Recommendation:** Reject

**Metareview:**

This paper studies RL with perturbed rewards, where a technical challenge is to revert the perturbation process so that the right policy is learned.  Some experiments are used to support the algorithm, which involves learning the reward perturbation process (the confusion matrix) using existing techniques from the supervised learning (and crowdsourcing) literature.

Reviewers found the problem setting new and worth investigating, but had concerns over the scope/significance of this work, mostly about how the confusion matrix is learned.  If this matrix is known, correcting reward perturbation is easy, and standard RL can be applied to the corrected rewards.  Specifically, the work seems to be limited in two substantial ways, both related to how the confusion matrix is learned.
  * The reward function needs to be deterministic.
  * Majority voting requires the number of states to be finite.
The significance of this work is therefore mostly limited to finite-state problems with deterministic reward, which is quite restricted.

As the authors pointed out, the paper uses discretization to turn a continuous state space into a finite one, which is how the experiment was done.  But discretization is likely not robust or efficient in many high-dimensional problems.

It should be noted that the setting studied here, together with a thorough treatment of an (even restricted) case, could make an interesting paper that inspires future work.  However, the exact problem setting is not completely clear in the paper, and the limitations of the technical contributions is also somewhat unclear.  The authors are strongly advised to revise the paper accordingly to make their contributions clearer.

Minor questions:
  - In lemma 2, what if C is not invertible.
  - The sampling oracle assumed in def 1 is not very practical, as opposed to what the paper claims.
  - There are more recent work at NIPS and STOC on attacking RL (including bandits) algorithms by manipulating the reward signals.  The authors may want to cite and discuss.